# Uncovering Language Model Processing Strategies with Non-Negative Per-Example Fisher Factorization

## Abstract

Understanding the heuristics and algorithms that comprise a model's behavior is important for safe and reliable deployment. While gradient clustering has been used for this purpose, gradients of a single log probability capture only a slice of the model's behavior, and clustering can only assign a single factor to each behavior. We introduce NPEFF (Non-Negative Per-Example Fisher Factorization), an interpretability method that overcomes these limitations by decomposing per-example Fisher matrices using a novel decomposition algorithm that learns a set of components represented by learned rank-1 positive semi-definite matrices. Through a combination of human evaluation and automated analysis, we demonstrate that these NPEFF components correspond to heuristics used by language models on a variety of text processing tasks. We find that NPEFF excels at decomposing behaviors comprised of multiple factors compared to the baselines of gradient clustering and activation sparse autoencoders. We also show how NPEFF can be adapted to be more efficient on tasks with few classes. We further show how to construct parameter perturbations from NPEFF components to selectively disrupt a given component's role in the model's processing. Along with conducting extensive ablation studies, we include experiments using NPEFF to study in-context learning.

## 1 Introduction

Transformer-based large language models (LLMs) have proven to be capable of a wide range of text-based tasks (Devlin, 2018; Achiam et al., 2023; Dubey et al., 2024; Yang et al., 2025). However, there is not yet a reliable means of understanding *why* a language model generated a given prediction. Towards this end, work on Transformer circuits aims to uncover interpretable computational graphs that underlie specific model behaviors such as indirect object identification, greater-than comparisons, and docstring completion (Elhage et al., 2021; Wang et al., 2022; Hanna et al., 2023; O'Neill & Bui, 2024; Hsu et al., 2024). While these circuits do appear to be directly related to model processing strategies, the behaviors under study must be specified ahead of time (via researcher intuition or expertise) instead of being uncovered in an unsupervised manner. This introduces bias and risks missing unintuitive model behaviors.

Recently, Michaud et al. (2023) and Marks et al. (2024) proposed clustering the gradient of the model's loss with respect to its parameters to unsupervisedly discover model behaviors. Intuitively, these methods conceptualize the model's processing as being comprised of a set of abstract internal modules. When a module is used to process an individual example, its corresponding parameters are reflected within the gradients of the loss. Under the strong assumption that a single module is used for each example – deemed "monogenic behaviors" by Michaud et al. (2023) – gradient clustering groups examples by the module used, and thus clusters correspond to model behaviors. However, model behaviors are likely to be *polygenic* in general, i.e. influenced by multiple factors from within the model. For example, contexts with ambiguous continuations will have at least one factor corresponding to each possibility. This mismatch has led to gradient clustering only being applied to contexts where the model correctly predicts the next token with low entropy (Michaud et al., 2023; Marks et al., 2024), leading to a heavy bias towards simple linguistic behaviors and precluding its application to studying more complex tasks. A more subtle issue comes from the use of loss gradients since they will capture only a narrow slice of the model's predictive distribution. This makes them poorly

suited for capturing information when, for example, a polygenic behavior consists of factors that influence predictions over different classes. Furthermore, this representation of model processing is biased towards factors influencing the specific class that the gradient is taken with respect to.

In this work, we introduce a method called NPEFF (non-negative per-example Fisher factorization) that is well suited for uncovering factors of polygenic behaviors. NPEFF uses the "per-example Fisher (PEF)" matrix to capture the model's processing for each example. The PEF matrix is a positive semi-definite (PSD) matrix that relates perturbations in parameters to changes in the model's predictive distribution (Fisher, 1922; Amari, 2016; Soen & Sun, 2021). A parameter perturbation will affect the model's behavior on an example if and only if it disrupts one or more of the internal modules used in its processing. Thus the internal modules used in an example's processing will get imprinted into the PEF. Unlike loss gradients, the PEF matrix takes into account the model's entire predictive distribution. NPEFF uses a novel decomposition algorithm to approximate PEFs as a non-negative combination of rank-1 PSD matrices. Thus NPEFF can directly represent polygenicity as a weighted combination of multiple factors influencing an example's prediction. We also introduce a cheaper variant of NPEFF called G-NPEFF that applies NPEFF's decomposition algorithm to PEF stand-ins constructed using the gradient of the log probability of the model's prediction. G-NPEFF performs similarly to NPEFF when the number of classes is small. As the number of classes grows, however, it misses polygenic factors and becomes biased towards dominant factors influencing the top predicted class.

While PEFs capture a meaningful notion of a prediction's sensitivity to different parameter values, their large size (quintillions of entries for modern-scale models with billions of parameters) makes them intractable to use. Analogous to Marks et al. (2024), we therefore use random projections (Achlioptas, 2003; Bingham & Mannila, 2001; Xie et al., 2017) to make representing PEFs tractable. Importantly, we demonstrate that it is possible to essentially "reverse" the random projection using methods from compressed sensing (Donoho, 2006; Candes & Tao, 2006; Tropp, 2006). This allow us to associate directions in parameter space to the behavioral factors that NPEFF uncovers. Perturbing the model parameters along these directions provides a means to selectively disrupt these factors and thus test whether they are genuinely used by the model.

After introducing methods that enable us to work efficiently with PEFs, we run NPEFF analysis on a variety of text processing tasks. We then explore properties of these decompositions through automated and human analysis while comparing to the baselines of gradient clustering and activation sparse autoencoders. We additionally demonstrate that our approach can recover parameter-space representations of behavioral factors, and we explore using NPEFF to analyze how in-context learning (ICL) can be explained in terms of behavioral factors of the zero-shot model. Finally, we conduct extensive ablation studies on NPEFF hyperparameters, exploring the impact of varying the number of components and demonstrating the validity of our PEF tooling.

## 2 Non-Negative Per-Example Fisher Factorization (NPEFF)

NPEFF consists of two main stages: computation of PEFs and decomposition over a set of PEFs. We use PEFs to capture per-example processing since they relate perturbations in parameters to perturbations over the model's entire predictive distribution. We use a combination of low rank representations and random projections to make the storage of PEFs over many examples tractable. The decomposition stage aims to represent the PEFs as a non-negative combination of rank-1 positive semi-definite (PSD) matrices, which ensures that the reconstructions are PSD like the PEFs themselves. By allowing multiple factors to be assigned to each PEF, this decomposition respects the polygenicity of the underlying behavioral factors.

### 2.1 Collection of PEFs

Consider the conditional distribution $p_\theta(y|\mathbf{x})$ produced by a model parameterized by $\theta \in \mathbb{R}^n$ given an example $\mathbf{x}$. We define the per-example Fisher (Fisher, 1922), or PEF, as the positive semidefinite (PSD) $n \times n$-matrix

$$F(\mathbf{x}) = \mathbb{E}_{y \sim p_\theta(y|\mathbf{x})} \nabla_\theta \log p_\theta(y|\mathbf{x}) \nabla_\theta \log p_\theta(y|\mathbf{x})^T. \tag{1}$$

The PEF allows us to relate small perturbations $\delta \in \mathbb{R}^n$ in the model parameters to changes in the model's predictive distribution $p_\theta(y|\mathbf{x})$ via (Amari, 2016):

$$D_{\mathrm{KL}}(p_\theta(y|\mathbf{x})\|p_{\theta+\delta}(y|\mathbf{x})) \approx \frac{1}{2}\delta^T F(\mathbf{x})\delta. \tag{2}$$

For a typical classification model, $p_\theta(y|\mathbf{x})$ corresponds to a categorical distribution over class labels. In this work, we additionally consider language models, which predict a sequence of token IDs from a vocabulary of tokens conditioned on a prefix. In this case, $p_\theta(y|\mathbf{x})$ is a distribution over a set of potential endings to a prefix $\mathbf{x}$, defined over a subset of the vocabulary, or the entire vocabulary. We consider any of these options to effectively define a set of possible predictions and continue to treat $p_\theta(y|\mathbf{x})$ as a categorical distribution (though we will consider a series of approximations to make calculation of the expectation in Equation (1) tractable in Section 2.1.1). Letting $r$ denote the number of categories, we can exactly represent the PEF using an $r \times n$-matrix $G(\mathbf{x})$ as $F(\mathbf{x}) = G(\mathbf{x})^T G(\mathbf{x})$ where the $i$-th row of $G(\mathbf{x})$ is equal to $\sqrt{p_\theta(y_i|\mathbf{x})}\nabla_\theta \log p_\theta(y_i|\mathbf{x})$. We call $G(\mathbf{x})$ the low-rank matrix representation of the PEF, or the LRM-PEF. Note that we do not make use of the fact that rows of the LRM-PEF $G(\mathbf{x})$ correspond to categories since we only interact with it via the PEF $F(\mathbf{x}) = G(\mathbf{x})^T G(\mathbf{x})$.

### 2.1.1  Approximating the Expectation

While we can conceptually consider the distribution over the set of possible continuations produced by a language model as a categorical distribution, handling the expectation over $y$ in equation 1 exactly is infeasible due to the large number of options (typically tens of thousands or more). While we could approximate the expectation by sampling multiple $y \sim p_\theta(y|\mathbf{x})$, we instead opt for a method based on random projections. This method allows for the PEF to capture information across the whole distribution of next-token probabilities.

Let $\mathbf{q}(\mathbf{x}; \theta) \in \mathbb{R}^r$ be defined element-wise with its $i$-th entry equal to $\mathtt{stop\_grad}(\sqrt{p_\theta(y_i|\mathbf{x})}) \log p_\theta(y_i|\mathbf{x})$, where $\mathtt{stop\_grad}$ treats a quantity as a constant while backpropagating. If $A \in \mathbb{R}^{r' \times r}$ is a random projection matrix (i.e. $A^T A \approx I$), then $\tilde{G}(\mathbf{x}) = \nabla_\theta A\,\mathbf{q}(\mathbf{x}; \theta) \in \mathbb{R}^{r' \times n}$ works well as a stand-in for the LRM-PEF $G(\mathbf{x})$ in the sense that $\tilde{G}(\mathbf{x})^T \tilde{G}(\mathbf{x}) \approx G(\mathbf{x})^T G(\mathbf{x}) = F(\mathbf{x})$. See Appendix B for a proof.

### 2.1.2  Rank Reduction

While the LRM-PEF representation $G \in \mathbb{R}^{r \times n}$ is exact, we can construct a lower-rank *approximation* $G' \in \mathbb{R}^{r' \times n}$, where $r' < r$, to the PEF using SVD to further reduce its storage costs by decreasing its number of rows. Notably, we can apply SVD after the random projection step (Section 2.2.1), which greatly reduces its computational cost. Consider the SVD $G = U\Sigma V^T$. Let $\Sigma \in \mathbb{R}^{r' \times r'}$ and $V' \in \mathbb{R}^{n \times r'}$ be the submatrices corresponding to the top $r'$ singular values. The reduced rank LRM-PEF is given by $G' = \Sigma' V'^T$. We can ignore the $U$ matrix since it is orthogonal and thus $U^T U = I$, so it does not affect the transformation from the LRM-PEF to the PEF matrix.

## 2.2  Decomposition

Given a set $G_1, \ldots, G_m$ of LRM-PEFs over a set of $m$ examples and $C$ components to learn, we define NPEFF as a non-negative factorization expressed as the non-convex optimization problem

$$\begin{aligned} \text{minimize} \quad & \sum_{i=1}^m \|G_i^T G_i - \sum_{j=1}^C W_{ij}\mathbf{h}_j\mathbf{h}_j^T\|_F^2 \\ \text{subject to} \quad & W_{ij} \geq 0. \end{aligned} \tag{3}$$

where $\mathbf{h}_j \in \mathbb{R}^n$ is the vector corresponding to component $j$, which we refer to as the component's "pseudo-Fisher". In words, NPEFF aims to find a set of $C$ rank-1 PSD matrices $\mathbf{h}_j\mathbf{h}_j^T$ and a set of non-negative coefficients $W_{ij}$ for each PEF that produces a good reconstruction in terms of Frobenius distance.

Our algorithm for efficiently solving equation 3 at scale is presented in Algorithm 1 with details in Appendix C. As a high level overview, we alternate between updating the coefficients $W_{ij}$ and updating the pseudo-Fisher vectors $\mathbf{h}_j$. The coefficient update is similar to a multiplicative update step in non-negative

---

**Algorithm 1** NPEFF decomposition

---

**Require:** LRM-PEFs $\{G_1, \ldots, G_m\} \subset \mathbb{R}^{r \times n}$, number of components $C \in \mathbb{N}$, learning rate $\eta > 0$, number of
   pseudo-Fisher only steps $N_1 \in \mathbb{N}$, number of joint steps $N_2 \in \mathbb{N}$
   **initialize** pseudo-Fisher vectors $H \in \mathbb{R}^{C \times n}$, coefficients $W \in \mathbb{R}^{m \times C}$ s.t. $W_{ij} > 0$
   **allocate** $B \in \mathbb{R}^{m \times r \times C}$, $N, D \in \mathbb{R}^{m \times C}$, $T_1, T_2 \in \mathbb{R}^{C \times n}$
   **for** $t = 1, \ldots, N_1 + N_2$ **do**
       $B_{ijk} \leftarrow \sum_{\ell=1}^{n} G_{ij\ell} H_{k\ell}$
       **if** $t \geq N_1$ **then**                                                                    ▷ Start of coefficient update step
           $N_{ik} \leftarrow \sum_{j=1}^{r} B_{ijk}^2$
           $D \leftarrow W((HH^T) \odot (HH^T))$
           $W_{ij} \leftarrow W_{ij} N_{ij} / D_{ij}$
       **end if**
       $T_1 \leftarrow 4((W^T W) \odot (HH^T))H$                                      ▷ Start of pseudo-Fisher update step
       $[T_2]_{i\ell} \leftarrow -4 \sum_{j=1}^{m} \sum_{k=1}^{r} W_{ji} B_{jki} G_{jk\ell}$
       $H \leftarrow H - \eta(T_1 + T_2)$
   **end for**
   **return** $H, W$

---

matrix factorization (Lee & Seung, 1999). The pseudo-Fisher update step is a gradient descent step. Notably, our implementation directly operates on the low-rank representations and does not require materializing any full PSD matrices.

We found it important for training stability to initialize the pseudo-Fisher vectors by performing only the pseudo-Fisher vector update step (i.e., skipping the coefficient update step) at the start of the optimization process. Furthermore, we normalize each PEF $G_i^T G_i$ to unit Frobenius norm. Failure to do this leads to the optimization loss being dominated by examples with large norm PEFs, which are usually atypical examples. Following completion of the decomposition, we normalize each component's rank-1 matrix to unit Frobenius norm and rescale the coefficients accordingly. This puts coefficients across components on the same scale. We finally note that given the pseudo-Fisher vectors of a decomposition, we can fit coefficients to an arbitrary (fixed) set of PEFs by only performing the coefficient update step repeatedly.

### 2.2.1 Random Projections

To make storing LRM-PEFs across a data set tractable, we apply a sparse random projection (Li et al., 2006) to each row of $G(\mathbf{x})$, thus reducing its number of *columns*. Note that this dimensionality reduction is separate to the SVD-based approach in Section 2.1.2 that reduces the number of *rows* of $G(\mathbf{x})$. Random projections are a dimensionality reduction procedure that approximately preserves inner products between vectors (Achlioptas, 2003; Vempala, 2005; Li et al., 2006). We developed a custom CUDA kernel that computes these projections efficiently without needing to materialize the projection matrix. See Appendix A for details.

In Appendix C.6, we provide a theoretical justification that the optimization problem equation 3 (and, consequently, our algorithm) is still meaningful when operating on randomly projected PEFs. We show that using projected PEFs should lead to approximately the same coefficients and pseudo-Fisher vectors as those that would have been produced if no random project was used. One drawback of operating on projected PEFs, however, is that the pseudo-Fisher vectors $\mathbf{h}_j$ belong to the projected space rather than the original parameter space. Parameter-space pseudo-Fisher vectors can be useful for confirming that NPEFF components do correspond to processing strategies used by the model. For example, consider perturbing the model parameters in the direction of the pseudo-Fisher vector. Following the information geometric interpretation equation 2 of PEF matrices, this should preferentially affect the model's predictions on examples with a large coefficient for that component.

When a vector is sparse, it can be possible to "reverse" a random projection and recover the projected vector from its projection via compressed sensing (Donoho, 2006). We have good reason to expect parameter-space

pseudo-Fisher vectors to be sparse because the overparameterization of models leads to gradients with respect many of the parameters being consistently insignificant (Frankle & Carbin, 2018). Since the pseudo-Fisher vector corresponds to the subset of parameters responsible for a particular behavioral factor, we can expect them to be even sparser.

We make use of the compressed sensing algorithm introduced in Hale et al. (2007). Given a random projection matrix $A \in \mathbb{R}^{n \times p}$ and projected pseudo-Fisher vector $\mathbf{h} \in \mathbb{R}^p$, this solves the compressed sensing problem $\arg\min_{\mathbf{u} \in \mathbb{R}^n} \|\mathbf{u}\|_1 + \frac{1}{2\mu}\|A\mathbf{u} - \mathbf{h}\|_2^2$ by starting with $\mathbf{u} = \mathbf{0}$ and iteratively setting $\mathbf{v} = \mathbf{u} - \tau A^t(A\mathbf{u} - \mathbf{h})$ and $\mathbf{u} \leftarrow \text{sign}(\mathbf{v})\max\{|\mathbf{v}| - \eta, 0\}$, where $|\cdot|$ is the element-wise absolute value and $\tau, \eta \in \mathbb{R}$ vary between steps. Notably, this algorithm only requires matrix-vector products with the random projection matrix and its transpose, which our custom CUDA kernels efficiently support. We leave development of more bespoke parameter-space pseudo-Fisher recovery algorithms to future work.

### 2.3 G-NPEFF

Although we have introduced methods for reducing the costs of computing and processing PEFs, they remain inherently more expensive than gradients. To explore the consequences of eliminating this overhead, we introduce G-NPEFF, which applies the NPEFF decomposition to rank-1 PEFs constructed using the gradient of the log-probability of the predicted class/token. More precisely, let $\mathbf{g}(\mathbf{x}) = \nabla_\theta \log p_\theta(y'|\mathbf{x})$, where $y' = \arg\max_y p_\theta(y|\mathbf{x})$, denote the gradient. G-NPEFF uses $\mathbf{g}^T(\mathbf{x})$ in-place of the LRM-PEF $G(\mathbf{x})$ when performing the decomposition in Section 2.2. Analogously to PEFs, we use random projections to make storage and handling of these gradients across many examples tractable, and we normalized the gradients such that $\mathbf{g}(\mathbf{x})\mathbf{g}^T(\mathbf{x})$ had unit Frobenius norm. G-NPEFF provides a cheaper alternative to full NPEFF since the gradients are cheaper to compute and store than the LRM-PEF estimates. It also allows us to explore the different information captured by PEFs and gradients without being confounded by a different decomposition algorithm. The gradients used by G-NPEFF coincide with the gradient of the loss on examples where the model happens to make the correct prediction, which was a restriction imposed by previous work using gradients to characterize model processing (Michaud et al., 2023; Marks et al., 2024). Like full NPEFF, this also allows G-NPEFF to not require ground truth labels.

## 3 Experiments

### 3.1 Characterizing Component Tunings

**Setup** To determine the types of behaviors uncovered by NPEFF and compare to baselines, we ran NPEFF and G-NPEFF on a representative group of models and tasks to explore the components they produced. We focus on language models and natural language tasks, but we expect that NPEFF would be effective in other modalities as well. We used the 360M parameter version of SmolLM2 (Allal et al., 2024) on the sentiment analysis task SST2 (Socher et al., 2013) with 2 classes, the topic identification task Yahoo Answers Topics (YAT) (Zhang et al., 2015) with 10 classes, the intent classification task CLINC150 (Larson et al., 2019) with 151 classes, and the open question answering task TriviaQA (Joshi et al., 2017). The $p_\theta(y|\mathbf{x})$ ranges over a set of suffixes for CLINC150, the entire vocabulary for TriviaQA, and a subset of the vocabulary for SST2 and YAT. We use a zero-shot formulation for the language models. See Appendix D for more information on the formulation of these tasks and how we constructed $p_\theta(y|\mathbf{x})$ for them.

For SST2, we used 60,000 examples, a projected dimension of 16,192, and 512 components. For YAT, we used 100,000 examples, a projected dimension of 16,192, and 2048 components. For CLINC150, we used 23,700 examples, approximated the expectation using 8 projections, SVD reduced the PEF rank to 4, a projected dimension of 16,192, and 512 components. For TriviaQA, we used 133,838 examples, a projected dimension of 8192, approximated the expectation using 16 projections, SVD reduced the PEF rank to 4, and used 2048 components. Analogous to Marks et al. (2024), we ignore the embedding and Layer Normalization (Zhang & Sennrich, 2019) parameters when computing the PEFs to focus on the processing done by the model's internals. We leave further exploration of parameter selection to future work. For all NPEFF decompositions, we used a warm-up of 1000 frozen coefficient steps with a learning rate of 1e-5 and another 3000 steps with a learning rate of 3e-4.

**Baselines** We compare NPEFF and G-NPEFF to two baselines: gradient clustering (Michaud et al., 2023) and activation SAEs (Gao et al., 2024). We do not include influence functions (Grosse et al., 2023) as a baseline since they explain model predictions in terms of influential training examples rather than behavioral factors.

Gradient clustering (GC) performs k-means clustering on gradients of the log-probability of the predicted class/token for each example, which are the same gradients used by G-NPEFF. Like the other methods, this enables GC to not require ground truth labels and coincides with the gradient of the loss when restricted to examples where the model makes the correct prediction as was done in Michaud et al. (2023). In all experiments, we applied the same random projection to the gradients as was applied to the PEFs. For the same reason as for PEFs, the gradients were normalized to unit L2 norm, and we ignored the embedding and LayerNorm parameters.

To adapt activations SAEs as a baseline method for uncovering model behaviors, we trained TopK-SAEs (Makhzani & Frey, 2013; Gao et al., 2024) over the task data using a single token's activation for each example. We take this activation from the output of the residual stream for the final token in the context. In all experiments, we used a value of $k = 32$ non-zero latents per example to control the sparsity. We also used the same total number of latents as the number of NPEFF components in the comparable experiments. Details on the training of these SAEs can be found in Appendix I.

Runtime comparisons between NPEFF, G-NPEFF, GC, and SAEs are provided in Appendix E. We include both time needed to compute PEFs/gradients/activations and the time needed for the decompositions.

### SST2
slightly disappointed
left slightly disappointed
disappointing to a certain degree
falls somewhat short

### YAT
How do i become slim in 3 months..i weigh 52 kg n im 5'1".?
how can i lose 10 lbs in a short amount of time?
how can i lose 30 pounds and it not take me a whole lot of time?
how can i lose 20 lbs in a hurry( i mean fast!) no drugs or supplements please?

### CLINC150
what does anachronistic mean
what does assiduous mean
what does circuitous mean
what's the meaning of a fortnight

### TriviaQA
In which year did Mozart die?
In which year did General Franco die?
In which year did Elvis die?
In which year did Beethoven die?

Figure 1: Top examples of selected components from NPEFF decompositions.

**Top component examples** Following existing work on interpretability methods that use sparse autoencoders to decompose activations (Rajamanoharan et al., 2024a), we begin to get a qualitative sense for a component's tuning by looking at the examples with the highest coefficient for each component. Top examples from a component selected from each of our NPEFF decompositions are presented in Figure 1. Top examples from random components can be found in Appendix J. These groups of top examples each have a clear theme relevant to the task and thus represent factors of the model's behavior. For example, the example component for YAT is most activated by questions about rapidly losing weight (generally indicating a "health" topic label).

To quantify these intuitions, we performed a human evaluation study, which was restricted to the NPEFF decomposition on TriviaQA due to cost. We created groups of examples that were either the top examples for a component or random examples (called "control"). Evaluators were asked to answer yes/maybe/no if the examples had a common theme and write a short description of the theme if present. Each group was seen by 2 different evaluators. We found that 79% of components had a detectable theme, determined by a yes or maybe label, with a false positive rate of only 3% on the control example groups. This analysis supports NPEFF components representing interpretable factors of behavior.

Table 1: Percentages of components exhibiting tunings across methods and tasks. The "LnP" metric means tuned to labels but not predictions. For each task, the highest LnP percentage is bold and second highest is underlined. This metric is not present for TriviaQA since it is an open-vocabulary question answering task without a fixed set of labels.

| Method | SST2 | | YAT | | CLINC150 | | TriviaQA |
|---|---|---|---|---|---|---|---|
| | Pred | LnP | Pred | LnP | Pred | LnP | Pred |
| NPEFF | 69.1 | **15.0** | 35.7 | **1.9** | 26.1 | **19.1** | 23.6 |
| G-NPEFF | 69.5 | 14.6 | 94.7 | 0.34 | 87.1 | 1.8 | 85.7 |
| GC | 100.0 | 0.0 | 99.9 | 0.0 | 96.9 | 0.0 | 75.5 |
| SAE | 10.2 | 0.98 | 6.3 | 1.0 | 3.5 | 6.3 | 2.1 |

**Verifying polygenicity**    Apart from manually inspecting top examples, we also hope to determine whether components have properties consistent with those of factors of polygenic behaviors. We can perform an automated analysis by considering a component as tuned if all of its top 16 examples had the same prediction or ground truth label, if present. Components with fewer than 16 examples were excluded from this analysis. For (G-)NPEFF, we rank examples in accordance to their component weighting. For gradient clustering, we rank based on the proximity to the cluster centroid. For SAE, we rank examples in descending order of their component coefficient.

Recall that polygenic behaviors are influenced by multiple factors. For polygenic behaviors, we would expect factors that frequently dominate behavior would be marked as prediction-tuned. However polygenic factors that rarely present by themselves might not be marked as such since their influence is likely to be countered by other factors on some of their top examples. Hence we would expect a significant fraction of both prediction-tuned and non-prediction-tuned factors. In contrast, we would expect all components to be prediction-tuned if they corresponded to monogenic factors since each example's prediction is the result of exactly one factor. While genuine factors can still be marked as not label-tuned if they correspond to flawed heuristics, label-tuned components by definition correspond to meaningful task-relevant factors. Overall, the presence of components that are label-tuned but not prediction-tuned (which we call "LnP-tuned") is the clearest single indicator of the recovery of genuine polygenic factors for tasks with a fixed set of labels.

Our results are presented in Table 1. We see that NPEFF's fraction of prediction-tuned components is most consistent with recovery of polygenic factors among all the methods with a significant fraction of both prediction-tuned and non-prediction-tuned components on all tasks. Furthermore, it recovers the largest fraction of verifiably polygenic factors for all tasks as measured by the LnP-tuned fraction.

The comparison between NPEFF and G-NPEFF highlights the additional information captured by PEFs over gradients especially as the number of classes grows. When the number of classes is small, gradients capture a significant slice of the model's behavior, so the difference between G-NPEFF and NPEFF is small. For tasks with many classes, however, the information captured by gradients becomes heavily biased towards the predicted class. This leaves out polygenic factors influencing predictions over other classes, and thus the decomposition becomes increasingly monogenic as indicated by the increasing fraction of prediction-tuned components and decreasing fraction of LnP-tuned components.

Gradient clustering captured almost entirely monogenic factors with most to all components being prediction-tuned and almost no verifiably polygenic factors recovered on any tasks. Since clustering assigns exactly one factor to each example, it essentially guarantees the recovery of only monogenic factors.

The tuning of SAE components followed a different pattern to the other methods, which can be explained by their use of activations to represent per-example processing. Since they are not computed using gradients of the model log probabilities like the other methods, they contain a significant portion of information irrelevant to the model's predictions. This makes them a more muddled representation of model behavior and leads to the low fraction of prediction-tuned components on all tasks. However, they still contain an unfiltered

Table 2: Perturbation results, where the values are the geometric mean of ratios across components. The largest KL ratio for each task is bold.

| Method | SST2 KL | SST2 Norm | YAT KL | YAT Norm | CLINC150 KL | CLINC150 Norm | TriviaQA KL | TriviaQA Norm |
|---|---|---|---|---|---|---|---|---|
| NPEFF | 16.5 | 0.82 | 22.0 | 0.94 | 0.70 | 0.83 | 0.88 | 0.84 |
| G-NPEFF | **19.9** | 0.82 | **25.5** | 0.90 | **15.5** | 0.88 | **51.1** | 0.90 |
| GC | 3.79 | 0.84 | 0.77 | 0.85 | 10.4 | 0.66 | 33.0 | 0.80 |

snapshot of the information influencing predictions unlike the gradients used by G-NPEFF and gradient clustering. Hence SAEs can uncover some verifiably polygenic factors even as the number of classes grows.

We ran further experiments using the 1.7B parameter version of SmolLM2 (Allal et al., 2024) on SST2 and YAT. The tuning results, presented in Appendix F, follow a similar pattern as for the 360M parameter model. Hence, NPEFF is well-suited for recovering polygenic factors across multiple model sizes.

## 3.2 Perturbations

Following the method described in Section 2.2.1, we can design perturbations using compressed sensing to reconstruct pseudo-Fisher vectors in parameter space from their projections. Instead of using the pseudo-Fisher vector directly as the perturbation, we can improve the selectivity of the perturbation's impact by orthogonally rejecting it from the other pseudo-Fisher vectors. This arises from wanting a perturbation orthogonal to all pseudo-Fisher vectors other than the one we want to affect. In practice, we found that only rejecting from vectors with an absolute cosine similarity less than a threshold worked best. More precisely, let $\mathbf{h}$ denote the pseudo-Fisher vector for the component we wish to perturb. Iterating over the components $i = 1, \ldots, C$, let $\hat{\mathbf{h}}_i$ be the L2-normalized pseudo-Fisher vector for the $i$-th component. If the absolute cosine similarity $|\mathbf{h}^T \hat{\mathbf{h}}_i|/\|\mathbf{h}\| < 0.5$, we replace $\mathbf{h}$ with the orthogonal rejection $\mathbf{h} - (\mathbf{h}^T \hat{\mathbf{h}}_i)\hat{\mathbf{h}}_i$. These similarity testing and orthogonal rejection steps can be performed using the projected vectors, i.e. before the compressed sensing step.

To evaluate the perturbed model, we are interested in measuring how much the predictions from a given component's top examples change. We therefore first compute the average KL-divergence of the perturbed model's predictions from the original model's predictions on a per-example basis and then report the ratio of the mean KL-divergence for the component's top examples to the mean KL-divergence over a set of random examples. We also report the ratios of the average PEF norms for these groups to represent how relatively sensitive top examples are to perturbations. This follows from equation 1, which relates parameter perturbations to the KL-divergence using the PEF, where we can see that increasing the scale of the PEF will increase the KL-divergence given a fixed perturbation. Since the rank-1 PSD matrix corresponding to a Fisher pseudo-vector is invariant under multiplying the vector by -1, we try perturbations in both directions and report the higher KL-ratio.

We ran perturbations experiments using the decompositions from Section 3.1. We used 128 randomly selected components for all tasks except for CLINC150, where we used 32 components. We used a similarity threshold of 0.5 for orthogonal rejection, used 16 component top examples, and a random set of 1,000 baseline examples except for CLINC150 where we used a random set of 200 baselines examples due to computational constraints. We used a perturbation L2 magnitude of 2e1. For the gradient clusters, we used the cluster centroids in place of the pseudo-Fishers. We restricted our analysis to clusters with at least 16 examples. We excluded SAEs since there is not an analogous way to use them to modify the model parameters.

Results are summarized in Table 2 with more experimental details presented in Appendix G. For most settings, the component top examples were significantly more affected by the perturbations than the random examples. This difference cannot be explained by component top examples simply being more sensitive to perturbations as indicated by the PEF norm ratios. These results indicate that the uncovered behavior

factors play a genuine role in the model behavior since we selectively disrupted the model's behavior on examples where a particular factor was deemed important.

The three failure cases were gradient clustering on YAT and NPEFF on CLINC150 and TriviaQA. Seeing as components from these decomposition had reasonable tunings, we suspect that these decompositions did not fail to capture behavioral factors. Instead, either the projected parameter space representations learned in the decompositions were of low quality, or the compressed sensing failed to accurately recover the original parameter space representations. In Appendix G.1, we report the per-example reconstruction loss and pseudo-Fisher vector absolute cosine-similarity distributions for all of the decompositions. None of the decompositions with failed perturbations had suspect values, which supports this assertion. For NPEFF, these issues might have arisen from the approximations needed to estimate the large rank PEFs. In contrast, the gradients used by G-NPEFF provide a narrower but less muddled snapshot of the model's processing.

Excluding these failure cases, we generally found G-NPEFF to produce the most selective perturbations with NPEFF being a bit less selective, which can be explained by G-NPEFF being biased towards recovering factors influencing the class with the highest probability. These factors are more likely to play a dominant role in the model's behavior on their top examples, and thus perturbing them will be more disruptive.

Perturbations from gradient clusters were significantly less selective than either NPEFF variant. This difference might be due to NPEFF's better handling of polygenic model behavior: Multiple factors would be imprinted into the gradients for each example under this hypothesis. While a single factor would dominate the examples in each cluster, other factors, especially correlated ones, would be present and thus contaminate the centroids. By contrast, NPEFF is free to disentangle the factors present in each example. Hence, the pseudo-Fisher for a component can be a purer representation of its corresponding factor.

### 3.3 Application – Analyzing ICL

In-context learning (ICL) involves including a prefix consisting of labeled examples when prompting a language model to perform a task. Some work has suggested that models often do not learn the rules of the task *per se* but rather learn the how the task is structured from the examples in the context (Min et al., 2022; Brown et al., 2020; Zhao et al., 2021; Liu et al., 2021a; Razeghi et al., 2022). Interestingly, for zero-shot SmolLM2-360M on SST2, we found that while many components were tuned to ground truth labels and the model's predictions, the ground truth and predicted label were often different. This indicates that the model's ability to distinguish between the classes is not being fully reflected in its predictions, further supporting a claim from Min et al. (2022) that the model has the capabilities to solve the zero-shot task but is not adapted to its specific formulation and label distribution.

Based on this, we constructed a linear classifier based on zero-shot NPEFF component tunings for SST2 from Section 3.1. Namely, we constructed a matrix $W \in \mathbb{R}^{2 \times C}$, where $C$ is the number of NPEFF components. We set $W_{ij}$ to 1 if the $j$-th component is tuned to the $i$-th ground truth label, which here means that 75% of its top 16 examples had $i$ as their ground truth label. We set $W_{ij}$ to 0 otherwise. Given the NPEFF coefficients $\mathbf{w}$ for a particular example, this classifier predicts $\arg\max W\mathbf{w}$ as the label. This classifier can be seen as forming predictions based on a weighted score of the factors that formed the model's prediction. This classifier can be seen as simulating the effects of ICL under the hypothesis that the model learns no new behaviors from the context and simply re-weights its existing behaviors to adapt to the task formulation.

We present our results for NPEFF and baselines in Table 3. The 6-shot ICL context was found using the 6 examples that produced the best performance on a subset of 1024 examples out of 50 randomly sampled sets of 6 examples, following Zhang et al. (2022). This context had an accuracy of 91.6% across the entire dataset compared to 63.8% for the zero-shot setting. For gradient clusters, we used a one-hot representation of clusters in place of the coefficients. Clusters with fewer than 16 examples were ignored.

The classifiers based on NPEFF and G-NPEFF coefficients were able to achieve accuracies comparable to ICL, and they made very similar predictions to the ICL set-up. Hence much of the performance gains of ICL can be explained by behaviors already present in the zero-shot set-up. This indicates, at least in this scenario, much of the gains from ICL come from adapting to the specific presentation of the task rather than the model learning new behaviors from the context. Comparatively, the classifier based on gradient clusters

Table 3: Accuracies and similarities (in terms of percentage of identical predictions) to the zero-shot and 6-shot ICL set-ups for the coefficients-based linear classifier for SmolLM2-360M on SST2. Values are percentages. The highest value in each column is bold.

| Method | Accuracy | Similarity Zero-Shot | Similarity ICL |
|--------|----------|---------------------|----------------|
| NPEFF | **88.7** | 69.2 | **89.0** |
| G-NPEFF | 88.6 | 69.0 | **89.0** |
| GC | 84.8 | 58.9 | 86.3 |
| SAE | 56.3 | **91.7** | 53.7 |

Table 4: Component tuning information in percentages for SST2 as we vary the dimension of the random projection for NPEFF with 512 components.

| $d_{\text{proj}}$ | 128 | 1024 | 8192 | 16,192 | 32,768 | 65,536 |
|------|-----|------|------|--------|--------|--------|
| Pred | 65.4 | 69.1 | 70.5 | 69.1 | 69.1 | 70.9 |
| LnP | 13.7 | 15.0 | 15.2 | 14.6 | 14.8 | 14.6 |

achieved a lower accuracy than the NPEFF variants and was less similar to both the zero shot and ICL set-ups. Again, this can be explained by NPEFF's better handling of polygenic behaviors. When multiple factors influence a model's prediction, gradient clustering is unable to disentangle them. Hence, the linear classifier will incorporate only the most dominant behavior of the zero-shot model. By contrast, NPEFF allows for a fine-grained view into the set of factors that the model can use in its predictions.

## 4  Ablations

We explored the impact of various hyperparameters of the PEF computation and NPEFF decomposition on the resultant components. Unless mentioned otherwise, the experimental settings were taken from Section 3.1.

**Random Projection Size**   We experimented with using random projection sizes of 128, 1024, 8192, 16,192, 32,768, and 65,536 for the SST2 NPEFF set-up from Section 3.1. Note that the 128 and 1024 sizes used a dense projection matrix since our implementation was significantly faster in those cases. Using the fractions of prediction-tuned and LnP-tuned components as a proxy for decomposition quality, we see from Table 4 all decompositions with a projected dimension of 1024 or greater performed similarly while the extremely small projected dimension of 128 only slightly deteriorated. The plateauing of decomposition quality with increasing projection dimension highlights that much of the information on model behavior is retained with even relatively aggressive projections.

**Number of Components**   We ran experiments varying the number of NPEFF components using set-ups similar to SST2 and CLINC150 from Section 3.1. We tried using 32, 64, 128, 256, and 512 components. We found that components tend to "split" as the number of components increase: Essentially, a component representing to a more general behavior gets converted to multiple components tuned to specific instantiations of that behavior. To map a component to its corresponding splits from another decomposition, we use the cosine similarity of component coefficient vectors to compare components between decompositions. For each component in the fine-grained decomposition, we find the coarse-grained component with the largest similarity score. This creates a map from each coarse-grained component to its set of corresponding fine-grained splits.

When running this identification, we find that all or almost all of the coarse-grained components have at least one matching fine-grained component among pairs of decompositions. Note that we would expected coarse-grained prediction-tuned components to have their prediction tuning preserved in their fine-grained

Table 5: Per-example held-in reconstruction loss (Recon-In), per-example held-out reconstruction loss (Recon-Out), and percentage of tuned components on the held-out set (Pred, LnP) for NPEFF as we vary the number of components for SST2 and CLINC150.

| | SST2 | | | | CLINC150 | | | |
|---|---|---|---|---|---|---|---|---|
| Components | Recon-In | Recon-Out | Pred | LnP | Recon-In | Recon-Out | Pred | LnP |
| 32 | 0.53 | 0.54 | 75.0 | 18.8 | 0.63 | 0.63 | 15.6 | 0.0 |
| 64 | 0.50 | 0.50 | 73.4 | 12.5 | 0.59 | 0.59 | 21.9 | 1.6 |
| 256 | 0.43 | 0.43 | 69.9 | 12.9 | 0.49 | 0.50 | 18.4 | 10.9 |
| 512 | 0.39 | 0.40 | 67.2 | 15.6 | 0.43 | 0.46 | 15.6 | 13.3 |
| 1024 | 0.36 | 0.38 | 66.9 | 14.5 | 0.38 | 0.42 | 13.9 | 11.5 |
| 2048 | 0.32 | 0.36 | 65.4 | 14.1 | 0.32 | 0.39 | 9.8 | 7.4 |

splits since they just represent more specific instantiations of a prediction-tuned behavior. For each pair of decompositions, we restrict our analysis to coarse-grained components with all of their top 16 examples having the same predicted label. We then count the number of fine-grained matches with the same prediction tuning and divide by the total number of matches to get the fraction of tunings preserved. We get a value of 85.9% and 46.0% averaged across all pairs of decompositions for SST2 and CLINC150, respectively, which indicates that tunings tend to carry over from the coarse-grained components to their splits. The lower value for CLINC150 might come from coarse-grained components actually being tuned to multiple predictions but biased towards a specific prediction. These would register as tuned to a single prediction but would have some fine-grained splits tuned to a different predictions. A full breakdown in provided in Appendix H.

To assist in creating a practical recipe for choosing the number of components, we ran further experiments on SST2 and CLINC150 using 32, 64, 128, 256, 512, 1024, and 2048 components. Unlike previous experiments, we learned an NPEFF decomposition using a held-in set and then fit coefficients to the pseudo-Fisher vectors on a held-out set. We used held-in/held-out sizes of 30,000/30,000 for SST2 and 12,000/11,700 for CLINC150. All other hyperparameters were taken from Section 3.1. We report per-example held-in reconstruction losses, per-example held-out reconstruction losses, and component tuning percentages on the held-out set in Table 5. Both the held-in and held-out losses continue decreasing as we increase the number of components; however, the gap between the two losses is increasing. This indicates that overfitting is not an issue at 2048 components though it would likely become an issue if the number of components were sufficiently large. The fraction of prediction-tuned components generally decreased as the number of components increased. The fraction of LnP-tuned components peaked at 512 components with CLINC150 demonstrating a sharper drop-off for values farther away from 512. We note, however, that the absolute number of prediction-tuned and LnP-tuned components was still increasing as the number of components increased.

Overall, we see that the desired granularity of the uncovered factors plays the biggest role in the choice of the number of components in the NPEFF decomposition. Even for a relatively high number of components such as around 10% of the number of examples for CLINC150, we do not observe substantial overfitting.

**Decomposition Random Seed**   We experimented with 3 different random seeds to initialize the coefficients and pseudo-Fisher vectors in the NPEFF decomposition. Our set-up was identical to SST2 in Section 3.1 and Section 3.2. We report the component tuning and perturbation experiment results in Table 6. Across the seeds, we see only a minor variation for the fractions of tuned components and the selectivity of perturbations. This indicates that the properties of an NPEFF decomposition is robust to the random seed used to initialize it.

**Expectation Approximation and SVD Rank Reduction**   Recall that to efficiently compute the expectation in Equation (1) when the space of possible outputs is large, we introduced a strategy using random projections. To experiment with the effect of this projection step, we ran experiments using SmolLM2-360M

Table 6: Tuning and perturbation results for SST2 as we vary the random seed of the NPEFF decomposition. For perturbation results, the values are the geometric mean across components.

|  | Tuning | | Perturbation | |
| --- | --- | --- | --- | --- |
| Seed | Pred | LnP | KL | Norm |
| 1 | 69.1 | 15.0 | 16.5 | 0.82 |
| 2 | 70.5 | 16.0 | 15.9 | 0.89 |
| 3 | 70.9 | 14.6 | 13.2 | 0.90 |

Figure 2: LRM-PEF SVD rank reduction on TriviaQA NPEFF decomposition. Each row represents SVD rank reductions for PEFs computed with a fixed EPS. Values are similarity scores to the full EPS rank decomposition within each row. Italicized values represent the similarity of two full rank EPS decompositions with different NPEFF random seeds.

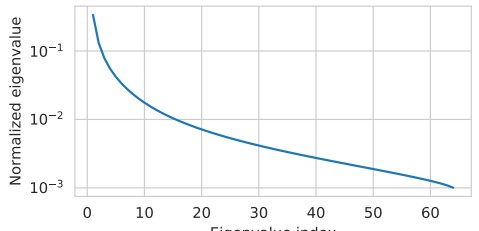

| | SVD rank | | | | | | |
| --- | --- | --- | --- | --- | --- | --- | --- |
| EPS | 1 | 2 | 4 | 8 | 16 | 32 | 64 |
| 64 | 0.75 | 0.83 | 0.87 | 0.90 | 0.91 | 0.94 | *0.89* |
| 16 | 0.75 | 0.82 | 0.87 | 0.89 | *0.89* | | |
| 4 | 0.74 | 0.84 | *0.86* | | | | |

Figure 3: Log-scale plot of the mean decay of normalized eigenvalues of PEF matrices with 64 expectation random projections.

on TriviaQA using 1, 4, 16, and 64 expectation projections. Further, recall that as discussed in Section 2.1.2, we use SVD to reduce the rank of the PEFs to reduce computational costs. To study the impact of using the SVD, we explore ranks of 1, 2, 4, 8, 16, 32, and 64, where applicable. We used a random projection size of 8192, used 40,000 examples, and 256 NPEFF components.

To create a similarity metric between a pair of NPEFF decompositions, we started with the cosine similarity of a pair of component coefficient vectors to compare components. For each component in one decomposition, we take the maximum cosine similarity with a component in the other decomposition. If this value is high for every component in both decomposition, then each component has a corresponding similar component in the other decomposition. We take the mean of this max cosine similarity across all components in the decompositions to get a single score.

Results for varying the SVD-reduced rank while holding the expectation projection size (EPS) fixed are provided in Figure 2. Each row contains the similarity score to the decomposition with the SVD rank set to the EPS. The last, italicized value in each row is the similarity between two full EPS rank decompositions from different NPEFF random seeds. Modest rank reduction has an effect on decomposition similarity close to that of changing the random seed, and even reducing the rank to 1 results in a fairly similar decomposition. We observe the general pattern of higher SVD-reduced ranks leading to decompositions more similar to the non-reduced rank case. We present a log-scale plot of the mean decay of normalized eigenvalues for the PEFs with expectation projection size 64 in Figure 3. Since the eigenvalues drop off quickly, we conclude that the SVD rank reduction can retain much of the information contained in the PEFs.

We also study the effects on the decomposition of varying the EPS while setting the SVD rank to the EPS. When compared to the EPS 64 decomposition, using an EPS of 1 produces a similarity score of 0.51, an EPS of 4 produces a similarity score of 0.66, and an EPS of 16 produces a similarity score of 0.79. This suggests that while using more projections in the expectation captures more information, even using a single projection can produce fairly similar decompositions. We leave further improvements, such as varying the number of projections used based on the entropy of the model's predictive distribution, to future work.

# 5 Related Work

**Fisher for ML**   The Fisher information matrix (FIM), which is the expectation of the PEF matrix equation 1 across the data set, has been used in machine learning for purposes including optimization (Osawa et al., 2023; Amari, 1998; Pascanu & Bengio, 2013; Osawa et al., 2020; Grosse & Martens, 2016; Martens & Grosse, 2015; Zhong et al., 2022; Tang et al., 2021), continual learning (Thompson et al., 2019; Kirkpatrick et al., 2017), compression (Chekalina et al., 2025; Pletenev et al., 2023; Theis et al., 2018; Hsu et al., 2022; Kwon et al., 2022; Liu et al., 2021b), merging (Matena & Raffel, 2022; Tam et al., 2023; Nathan et al., 2024; Lee et al., 2025), federated learning (Jhunjhunwala et al., 2024), task embeddings (Ma et al., 2023; Achille et al., 2019), and analysis (Hannun et al., 2021; Farokhi & Sandberg, 2017; Arnold et al., 2023; Achille et al., 2017). Frequently only its diagonal is used for the sake of tractability (Soen & Sun, 2024; Kirkpatrick et al., 2017) although other approximations have been used (Koroko et al., 2022; Martens & Grosse, 2015; Martens, 2020; Grosse & Martens, 2016; Chekalina et al., 2025). The empirical FIM, which uses only the gradient of the ground truth label instead of computing the expectation over the model's predictive distribution, is often used in place of the actual FIM; however, the empirical FIM suffers from several limitations (Wu et al., 2024; Martens, 2020; Kunstner et al., 2019; Thomas et al., 2020). Our use of the per-example FIM to characterize the model's per-example processing is novel though we note that Fisher kernels use the score function $\nabla \log p_\theta(\mathbf{x})$ of a generative model to produce per-example representations with similarity being computed via a kernel with the inverse of the FIM (Jaakkola & Haussler, 1998; Perronnin et al., 2010; Sánchez et al., 2013; Saunders et al., 2002; Holub et al., 2005; Van Der Maaten, 2011).

**Tensor Decompositions**   The NPEFF decomposition problem equation 3 can be phrased as a tensor decomposition problem (Kolda & Bader, 2009). Namely, we wish to represent the third-order tensor of stacked PEFs as a sum of rank-1 tensors, which is a variant of INDSCAL decomposition (Husson & Pagès, 2006; Carroll & Chang, 1970; Stegeman et al., 2006; Dosse et al., 2011). While INDSCAL is usually solved using a more general CP decomposition algorithm (Carroll & Chang, 1970; Harshman et al., 1970; Faber et al., 2003; Tomasi & Bro, 2006), our algorithm is more similar to a multiplicative update algorithm for non-negative matrix factorization (Lee & Seung, 1999; Burred, 2014; Boureima et al., 2024), where positive semi-definiteness takes that place of non-negativity for one of the factors and gradient descent is used to update it.

**Interpretability**   Various interpretability methods have used gradients to determine which input features most influence the predictions for a single example (Simonyan et al., 2013; Smilkov et al., 2017; Sundararajan et al., 2017). Unlike our use of gradients with respect to model parameters, these methods use gradients with respect to input features to solve the feature attribution problem.

Sparse autoencoders (SAEs) are used to learn an overcomplete representation of activations with a sparsity-inducing prior or function on the latents (Bricken et al., 2023a; Cunningham et al., 2023; Gao et al., 2024; Rajamanoharan et al., 2024b;a). SAE latents have been found to be more monosemantic and human interpretable than other features such as individual neurons (Lieberum et al., 2024; Lawson et al., 2024; Braun et al., 2024; Kissane et al., 2024; Templeton et al., 2024b; Paulo et al., 2024; Balcells et al., 2024; Lan et al., 2024; Brinkmann et al., 2025). Transcoders learn the input-output mapping of MLP layers with a sparsity-inducing prior on a larger hidden dimension (Dunefsky et al., 2024; Templeton et al., 2024a). Jacobian SAEs learn a pair of SAEs for the inputs and outputs of an MLP with a sparsity inducing prior on their Jacobians (Farnik et al., 2025).

Research in transformers circuits aims to find a computational circuit responsible for a particular behavior (Elhage et al., 2021). These behaviors are typically simple linguistic behaviors that include indirect object identification, greater-than comparisons, and docstring completion (Wang et al., 2022; Hanna et al., 2023; O'Neill & Bui, 2024; Hsu et al., 2024). While typically specified ahead of time, Marks et al. (2024) uses gradient clustering to unsupervisedly find behaviors. Some other works aim to explain the model's global behavior that was trained on a synthetic task (He et al., 2024; Nanda et al., 2023). The transformer is then represented as a computational graph where the granularity of nodes varies based on the work and includes MLPs (Hanna et al., 2023), attention heads (Olsson et al., 2022; Wang et al., 2022), SAE features (Marks et al., 2024; O'Neill & Bui, 2024; He et al., 2024), and transcoder features (Dunefsky et al., 2024). The

graph is then pruned to remove nodes and edges that are unimportant to the behavior using methods that include greedy patching (Conmy et al., 2023), first-order estimates of importance (Syed et al., 2023; Hanna et al., 2024), and a learned mask (Bhaskar et al., 2024).

Influence functions can be used to explain a model's behavior on particular example in terms of influential training examples by approximating the effect of adding or removing training samples on the parameters (Hampel, 1974; Grosse et al., 2023). This is accomplished via inverse Hessian vector products (Koh & Liang, 2017) or inverse Gauss-Newton Hessian vector products (Bae et al., 2022). Although the Guass-Newton Hessian matrix coincides with the Fisher information matrix for transformer language models (Martens, 2020), NPEFF differs from influence functions by decomposing per-example Fisher matrices and by explaining behaviors directly in terms of directions in parameter space.

## 6  Discussion

**Comparison to existing methods**  The two main novelties introduced by NPEFF are using PEFs to characterize per-example processing and the decomposition via non-negative coefficients over rank-1 PSD matrices. Compared to gradient clustering (Michaud et al., 2023), NPEFF better captures the ground truth where multiple factors influence the model's behavior on any particular example. Activation SAEs (Gao et al., 2024) provide an alternative view on model internals since activations capture what information is present at a particular location within the model but do a poor job at capturing the computation. Influence functions (Grosse et al., 2023) explain behavior in terms of influential training examples. Hence they cannot find behavioral factors like NPEFF. The main disadvantage of NPEFF the computational overhead associated with PEFs. We explored mitigating this by running NPEFF's decomposition on gradients for G-NPEFF, but we found that G-NPEFF performed poorly at uncovering polygenic components as the number of classes increased.

**Improving language models**  Though we have mostly focused in this work on introducing and evaluating NPEFF as an interpretability method, the potential exists to use it to improve language models. Examining component tunings could uncover beneficial or problematic behaviors. We also provide a means to produce the directions in parameter space corresponding to particular behaviors. These could be used as in our perturbation experiments to selectively disrupt particular behaviors. We leave development of more sophisticated methods to modify model behavior based on NPEFF to future work.

## 7  Conclusion

NPEFF represents a novel method that is well-suited to uncovering factors of polygenic model behaviors. We introduced PEFs as a novel object to characterize a model's processing of an example along with tools to make working with them tractable. Examining properties of NPEFF components, we saw that they corresponded to interpretable factors of behavior. Furthermore, we demonstrated NPEFF uncovered more factors of polygenic behavior compared to the baselines of gradient clustering and activations SAEs. NPEFF's decomposition can be applied to gradients with the cheaper G-NPEFF method, which that produces similar results when the number of classes is small. However when the number of classes is large, G-NPEFF focuses on dominant factors and recovers fewer polygenic factors. Using compressed sensing to construct parameter perturbations from projected component representations, we selectively disrupted their associated behaviors. In addition to conducting extensive ablation studies, we used NPEFF to analyze ICL. In future work, we aim to explore more refined evaluation metrics to aid in the comparison of methods along with better approximations for PEFs over a large number of classes.

## Broader Impact Statement

NPEFF provides a means to decompose the behavior of a language model into factors and a means to relate those factors to directions in parameter space. This could enable directed modification of model behaviors. While this can be beneficial if positive behaviors are amplified and negative behaviors are suppressed, this

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

# A    Random Projections

**Random projection matrix form**    To project from $\mathbb{R}^n$ to $\mathbb{R}^r$, let $A \in \mathbb{R}^{n \times r}$ be a random projection matrix. For computational purposes, we want most entries of $A$ to be zero. To do this, let $\mathbf{a} \in \mathbb{R}^n$ denote a column of $A$. We pick a region size $s \in \mathbb{N}$ and divide the entries of $\mathbf{a}$ into chunks of size $s$. In each of these chunks, we pick an entry at random and set it with equal probability to either 1 or -1. All other entries are 0. Following Marks et al. (2024), we pick a level of sparsity such that each entry of the original vector will, on average, contribute to 32 entries in its projection. This corresponds to selecting a sparse region size of $s = r/32$.

**Efficient projections using a CUDA kernel**    Note that explicitly materializing even a sparse representation of the projection matrix would take $32n$ values, i.e. 32 times the number of model parameters. However, we can construct entries of the matrix on the fly using a pseudo-random number generator. Hence the seed of the pseudo-random number generator essentially parameterizes the projection matrix.

This is a good fit to implement the projection via a CUDA kernel. In our current implementation, each thread in the kernel corresponds to one of the $r$ entries in the projection. The global seed specifying the projection matrix is used to produce a different seed for each thread. This pseudo-random number generator is then used to select the entry from each sparse region and its value.

# B    Expectation Approximation Proof

Let us start by restating the expressions from Section 2.1.1. Let $\mathbf{q}(\mathbf{x}; \theta) \in \mathbb{R}^r$ be defined element-wise with its $i$-th entry equal to $\texttt{stop\_grad}(\sqrt{p_\theta(y_i|\mathbf{x})}) \log p_\theta(y_i|\mathbf{x})$, where $\texttt{stop\_grad}$ treats a quantity as a constant while backpropagating. Note that $\nabla_\theta \mathbf{q}(\mathbf{x}; \theta) = G(\mathbf{x}) \in \mathbb{R}^{r \times n}$ is the LRM-PEF.

Let $A \in \mathbb{R}^{r' \times r}$ be a random projection matrix (i.e. $A^T A \approx I$). Let $G'(\mathbf{x}) = \nabla_\theta A \mathbf{q}(\mathbf{x}; \theta) \in \mathbb{R}^{r' \times n}$ be our approximation to the LRM-PEF. We have $G'(\mathbf{x}) = \nabla_\theta A \mathbf{q}(\mathbf{x}; \theta) = A \nabla_\theta \mathbf{q}(\mathbf{x}; \theta) = AG(\mathbf{x})$. When constructing the full PEF, we have $F'(\mathbf{x}) = G'(\mathbf{x})^T G'(\mathbf{x}) = G(\mathbf{x})^T A^T A G(\mathbf{x}) \approx G(\mathbf{x})^T I G(\mathbf{x}) = G(\mathbf{x})^T G(\mathbf{x}) = F(\mathbf{x})$. Hence our use of this random projection to approximate the expectation results in a similar PEF to representing the full expectation.

# C    Decomposition Algorithm

We start by reviewing the optimization problem from Section 2.2. Given a set $G_1, \ldots, G_m$ of LRM-PEFs over a set of $m$ examples and a number $C$ of components to learn, NPEFF can be expressed as the non-convex optimization problem

$$
\begin{aligned}
\text{minimize} \quad & \sum_{i=1}^m \|G_i^T G_i - \sum_{j=1}^C W_{ij} \mathbf{h}_j \mathbf{h}_j^T\|_F^2 \\
\text{subject to} \quad & W_{ij} \geq 0.
\end{aligned}
\tag{4}
$$

We stack the PEFs into a single 3-dim tensor $G \in \mathbb{R}^{m \times r \times n}$. We can express the quantities to be learned via the matrices $W \in \mathbb{R}^{m \times C}$ and $H \in \mathbb{R}^{C \times n}$, where rows of $H$ correspond to the $\mathbf{h}_j$. Our optimization of equation 4 proceeds in alternating steps of updating the coefficients $W$ and pseudo-Fishers $H$. Our $W$-update step is essentially a the $W$-update step from the multiplicative update NMF (Lee & Seung, 1999). algorithm. For the $H$-update step, we perform a gradient descent step with a fixed learning rate.

## C.1    $W$-**Update Step**

Recall that the multiplicative update step in NMF involves computing non-negative numerator and denominator matrices $N, D \in \mathbb{R}^{m \times C}$. The matrix $W$ is then updated via the element-wise rule $W_{ij} \mapsto W_{ij} N_{ij}/D_{ij}$.

Computing the numerator starts with computing the 3-dim tensor $B \in \mathbb{R}^{m \times r \times C}$, where $r$ is the rank used to represent the LRM-PEFs, with elements given by $B_{ijk} = \sum_{\ell=1}^n G_{ij\ell} H_{k\ell}$. The numerator is then given element-wise by $N_{ik} = \sum_{j=1}^r B_{ijk}^2$. The denominator is then given by $D = W((HH^T) \odot (HH^T))$, where $\odot$ denotes the Hadamard product.

## C.2   $H$-Update Step

The gradient of the loss with respect to $H$ consists of two terms $T_1, T_2 \in \mathbb{R}^{C \times n}$ that are added together. The first term is given by $T_1 = 4((W^T W) \odot (HH^T))H$. Computation of the second term starts by computing the 3-dim tensor $B \in \mathbb{R}^{m \times r \times C}$ as was done for the $W$-update step. The second term is then obtained element-wise as $[T_2]_{i\ell} = -4 \sum_{j=1}^{m} \sum_{k=1}^{r} W_{ji} B_{jki} G_{jk\ell}$.

## C.3   Multi-GPU Implementation Details

To speed up decompositions and support larger decompositions, we implement a multi-GPU strategy for our algorithm. We partition the input PEFs and the coefficients $W$ along the batch dimension across separate GPUs. We replicate the pseudo-Fisher matrix $H$ on each GPU. The $W$-update step can proceed on each GPU without the need for inter-GPU communication. Since the gradient of the loss with respect to $H$ can be expressed as a sum of per-example gradients, we compute its gradient for the samples residing on each GPU. Then a single all-reduce step is needed to aggregate the gradients for all of the examples. Then each GPU applies the gradient descent step to their own local copies of $H$.

## C.4   Other Considerations

We initialized $W$ using the uniform distribution on $[0, 1]$. We initialized $H$ using a normal distribution with zero mean and standard deviation of $\sqrt{2}/\sqrt{Cn}$. Since the PEFs were normalized to unit Frobenius norms, we chose this scaling so that the initial reconstructions would also have roughly unit Frobenius norms as well.

After initialization, we found it crucial to freeze $W$ and only train $H$ for a bit before commencing joint training. This is because if the $H$ is a poor fit for the $W$, the $W$ update step will end up setting $W$ to zero. Since the $W$ update is multiplicative, it remains zero throughout the remainder of training if this happens. We suspect that this behavior can be explained due to the nature of the multiplicative update step. It can be shown that the multiplicative update step is equivalent to gradient descent with a variable element-wise learning rate (Burred, 2014). Unlike traditional gradient descent that uses a small gradient step, the variable learning can become large. This makes it possible for the $W$ to jump directly to zero or some similarly small value. If the loss is greater than the Frobenius norm of the PEFs, then setting $W$ to zero will result in a lower loss. Hence, jumping to zero can decrease the loss in such cases.

## C.5   Convergence

While we do not provide a proof of convergence of our NPEFF decomposition algorithm, we can make a heuristic argument for its convergence. Following the proof of convergence for regular multiplicative-update NMF (Lee & Seung, 1999), we can show that the loss will be non-increasing following the $W$-update step. For a sufficiently small step size, we can expect the gradient descent step from the $H$-update to not increase the loss as well. Since the loss is bounded from below by 0, it follows that the loss should eventually converge. When actually running NPEFF, we found the loss to be non-increasing with the rate of decrease decelerating as the number of steps increased.

In practice, we did not encounter any issues with convergence given a long enough $H$-only update stage and a sufficiently low learning rate.

## C.6   Decomposition on Randomly Projected PEFs

Let us see consider the difference between an update step in the original and non-projected set-ups. Let $A \in \mathbb{R}^{n \times p}$ denote the random projection matrix used. While random projections only approximately preserve inner products, i.e. $AA^T \approx I$, we will make that assumption that they preserve inner products exactly, i.e. $AA^T = I$, to show that our algorithm commutes with random projections under that assumption. Let $G'_i = G_i A$ and $H' = HA$ denote the projected PEFs and pseudo-Fisher matrices, respectively. Note that the random projection will not directly affect the coefficients $W$.

Let's first look at the $W$-update step. Computing the 3-dim tensor $B$ is equivalent to computing $G_i H^T$ for $i = 1, \ldots, m$. We have $G_i' H'^T = G_i A A^T H^T = G_i H^T$, so $B$ is left unchanged when using the projections. The numerator $N$ is purely a function of $B$, so it is unchanged as well. The denominator $D$ depends on projected quantities solely through $H H^T$. Since $H' H'^T = H A A^T H^T = H H^T$, the denominator is unaffected by the use of the projection. Since both $N, D$ are unaffected by the projection, the $W$-update step is the same regardless of whether we using random projections.

Now let's look at the $H$-update step. Since we have shown that $H' H'^T = H H^T$, it follows that $T_1' = T_1 A$. Similarly since the random projection does not affect $B$, we can show that $T_2' = T_2 A$. Let $\eta > 0$ denote the learning rate used in the $H$-update step. Originally, the $H$-update step proceeds as $H \mapsto H - \eta(T_1 + T_2)$. With the random projection, the updated $H'$ is provided by $H' - \eta(T_1' + T_2') = (H - \eta(T_1 + T_2))A$. Essentially, we have shown that the $H$-update step commutes with random projections; performing an $H$-update in the original space followed by a random projection is equivalent to performing the projections first followed by the update step on the projections.

We have just shown that both the $W$-update and $H$-update leave the relationship between original and projected quantities unchanged: $W' = W$ and $H' = HA$. Thus by induction, these relationships hold after an arbitrary number of update steps. Hence operating on projected PEFs will produce the same coefficients as operating on original PEFs. Furthermore, the final pseudo-Fisher vectors from the projected decomposition will be equal to the projections of the final pseudo-Fisher vectors from the original decomposition.

One major caveat of this analysis, however, is that it assumes that the random projections preserve inner products exactly. In reality, inner products are only approximately preserved by the projections. Hence this analysis should only be expected to hold approximately in practice.

## D  Task Formulations

### D.1  SST2

SST2 consists of sentiment analysis of a sentence. Given a sentence, we construct a prompt via the template `Review: {sentence}\nSentiment:`. The 2 labels used for this task are `Negative` and `Positive`. To get a distribution $p_\theta'(y|\mathbf{x})$ over two labels, we start with the full model distribution $p_\theta(y|\mathbf{x})$ over the entire vocabulary given the context. We look at the first token in the tokenization of each label[1] and obtain their corresponding probabilities from $p_\theta(y|\mathbf{x})$. These two probabilities are then normalized to get a distribution $p_\theta'(y|\mathbf{x})$ over the labels. As a practical note, we can obtain the logits of $p_\theta'(y|\mathbf{x})$ by simply selecting the logits corresponding to label tokens from $p_\theta(y|\mathbf{x})$ due to properties of the softmax function.

### D.2  YAT

We phrased YAT as determining the topic corresponding to a question. Given a question, we construct a prompt via the template `Question: {question}\nWhat broad topic is this question about?  Choose from:\nSociety & Culture\nScience & Mathematics\nHealth\nEducation & Reference\nComputers & Internet\nSports\nBusiness & Finance\nEntertainment & Music\nFamily & Relationships\nPolitics & Government.\nTopic:`. The 10 labels used for this task are `Society & Culture`, `Science & Mathematics`, `Health`, `Education & Reference`, `Computers & Internet`, `Sports`, `Business & Finance`, `Entertainment & Music`, `Family & Relationships`, and `Politics & Government`. We construct a distribution $p_\theta'(y|\mathbf{x})$ over these 10 labels from the full model distribution $p_\theta(y|\mathbf{x})$ using the same process we used for SST2.

### D.3  CLINC150

CLINC150 consists of inferring the intent from one of 151 options given a query. Given a query, we construct a prompt via the template `Query: {query}\nIntent:`. The 151 labels are

---

[1]The SmolLM2 tokenizer prepends the space to the start of tokens, so our labels technically have a space at the start for all of the tasks.

Table 7: Approximate time to compute a characterization of a single example's processing in seconds for SmolLM2-360M.

| Object | SST2 | YAT | CLINC150 | TriviaQA |
|--------|------|-----|----------|----------|
| PEF | 0.49 | 3.5 | 8.0 | 2.3 |
| Gradient | 0.39 | 0.40 | 1.6 | 0.69 |
| Activation | 3.6E-3 | 1.3E-2 | 3.4E-3 | 1.4E-2 |

oos, freeze account, routing, pin change, bill due, pay bill, account blocked, interest rate, min payment, bill balance, transfer, order checks, balance, spending history, transactions, report fraud, replacement card duration, expiration date, damaged card, improve credit score, report lost card, card declined, credit limit change, apr, redeem rewards, credit limit, rewards balance, application status, credit score, new card, international fees, food last, confirm reservation, how busy, ingredients list, calories, nutrition info, recipe, restaurant reviews, restaurant reservation, meal suggestion, restaurant suggestion, cancel reservation, ingredient substitution, cook time, accept reservations, what song, play music, todo list update, reminder, reminder update, calendar update, order status, update playlist, shopping list, calendar, next song, order, todo list, shopping list update, smart home, current location, oil change when, oil change how, uber, traffic, tire pressure, schedule maintenance, gas, mpg, distance, directions, last maintenance, gas type, tire change, jump start, plug type, travel notification, translate, flight status, international visa, timezone, exchange rate, travel suggestion, travel alert, vaccines, lost luggage, book flight, book hotel, carry on, car rental, weather, alarm, date, find phone, share location, timer, make call, calculator, definition, measurement conversion, flip coin, spelling, time, roll dice, text, pto request status, next holiday, insurance change, insurance, meeting schedule, payday, taxes, income, rollover 401k, pto balance, pto request, w2, schedule meeting, direct deposit, pto used, who made you, meaning of life, who do you work for, do you have pets, what are your hobbies, fun fact, what is your name, where are you from, goodbye, thank you, greeting, tell joke, are you a bot, how old are you, what can i ask you, change speed, user name, whisper mode, yes, change volume, no, change language, repeat, change accent, cancel, sync device, change user name, change ai name, reset settings, and maybe.

To get a distribution $p'_\theta(y|\mathbf{x})$ over these 151 labels, we start by computing the probability of each label as a suffix to the query. These probabilities are then normalized to get a distribution $p'_\theta(y|\mathbf{x})$ over the labels. As a practical note, we can obtain the logits of $p'_\theta(y|\mathbf{x})$ by simply taking the log probability of each suffix due to properties of the softmax function.

### D.4 TriviaQA

TriviaQA is an open-form question answering task where the model generates an answer given a question. Given the question, we construct a prompt via the template `Question: {question}\nAnswer:`. We take $p_\theta(y|\mathbf{x})$ to be the model's next-token predictive distribution given this context.

## E  Runtime Information

We report the approximate times for the experiments in Section 3.1 if they were run using a single A6000 GPU. However, we note that the computation of per-example information can be trivially parallelized across examples. Furthermore, our NPEFF decomposition implementation supports parallelization across multiple GPUs to obtain a speed up. The timing information for computation of PEFs, gradients, and activations is provided in Table 7. The timing information to perform decompositions is provided in Table 8.

Table 8: Approximate times for computing the decompositions on a single A6000 GPU. The value in parantheses for the NPEFF variants is the step time for the $H$-only update.

| | SST2 | | YAT | | CLINC150 | | TriviaQA | |
| --- | --- | --- | --- | --- | --- | --- | --- | --- |
| Method | Step (ms) | Total (s) | Step (ms) | Total (s) | Step (ms) | Total (s) | Step (ms) | Total (s) |
| NPEFF | 310 (260) | 1190 | 3600 (3150) | 13,950 | 205 (185) | 800 | 2940 (2900) | 11,220 |
| G-NPEFF | 235 (180) | 885 | 1810 (1370) | 6800 | 85 (63) | 318 | 2090 (1520) | 7790 |
| GC | 85 | 5 | 635 | 47 | 38 | 1 | 467 | 14 |
| SAE | 110 | 5690 | 110 | 5690 | 130 | 6830 | 110 | 5690 |

Table 9: Percentages of components exhibiting tunings across methods and tasks for SmolLM2-1.7B. The "LnP" metric means tuned to labels but not predictions. For each task, the highest LnP percentage is bold and second highest is underlined.

| | SST2 | | YAT | |
| --- | --- | --- | --- | --- |
| Method | Pred | LnP | Pred | LnP |
| NPEFF | 65.2 | **20.9** | 20.9 | **1.4** |
| G-NPEFF | 64.8 | 20.1 | 89.8 | 0.1 |
| GC | 100.0 | 0.0 | 99.7 | 0.0 |
| SAE | 6.8 | 0.8 | 2.5 | 0.9 |

# F   Experiments on SmolLM2-1.7B

To see if NPEFF's suitability for recovering polygenic factors extended beyond the SmolLM2-360M model used in Section 3.1, we ran additional tuning experiments using SmolLM2-1.7B (Allal et al., 2024), which was 1.7B parameters. We restricted our analysis to SST2 and YAT, using the same hyperparameters as used for the corresponding experiments in Section 3.1. We present our results in Table 9. Similarly to the main text, we see that NPEFF's tuning results are most consistent with recovery of polygenic factors. It recovers a significant fraction of both prediction-tuned and non-prediction-tuned factors. Furthermore, it uncovers the highest fraction of LnP-tuned components.

# G   Perturbation Experimental Details

Compressed sensing aims to solve the constrained optimization problem

$$\min_{\mathbf{x}} \left\{ \|\mathbf{x}\|_1 : A\mathbf{x} = \mathbf{b} \right\}, \tag{5}$$

where $\mathbf{x} \in \mathbb{R}^n$ is the reconstruction which we wish to find, $\mathbf{b} \in \mathbb{R}^p$ is the projected vector, and $A \in \mathbb{R}^{p \times n}$ is the random projection matrix. However, we use the unconstrained relaxation

$$\min_{\mathbf{x}} \|\mathbf{x}\|_1 + \frac{1}{2\mu} \|A\mathbf{x} - \mathbf{f}\|_2^2, \tag{6}$$

where $\mu > 0$ is a hyperparameter controlling sparsity of the solution.

We used the fixed point continuation (FPC) solver from Hale et al. (2007) to solve equation 6 since it only requires matrix-vector products with the random projection matrix and its transpose. This is a good fit for our custom CUDA kernels implementing the random projections. We use the default hyperparameter values of $\gamma = 0.99$, $\beta = 4$, and equation (73) from Hale et al. (2007) to set $\tau$. We used a value of 4e-5 for the hyperparameter $\mu$. We also use a single inner iteration since we found that to produce the best results in

Table 10: Per-example reconstruction losses (i.e. per-example squared Frobenius/L2 distance) for the decompositions used in the perturbation experiments in Section 3.2. Entries corresponding to failed perturbation experiments are in bold.

| Method | SST2 | YAT | CLINC150 | TriviaQA |
|---|---|---|---|---|
| NPEFF | 0.40 | 0.33 | **0.44** | **0.75** |
| G-NPEFF | 0.40 | 0.35 | 0.48 | 0.66 |
| GC | 0.32 | **0.28** | 0.37 | 0.53 |

Table 11: Histograms of absolute cosine-similarities of pseudo-Fisher vectors for the decompositions used in the perturbation experiments in Section 3.2. Since the absolute cosine-similarity is bounded between 0 and 1, all x-axes span from 0 to 1.

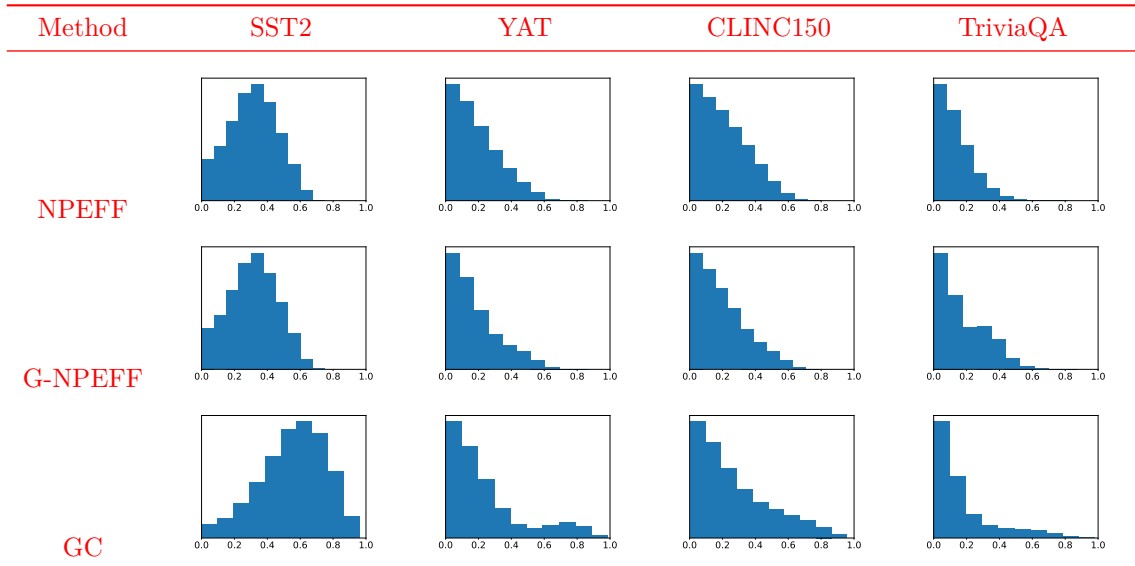

our perturbation experiments. We found this algorithm to produce reconstructions quickly, typically taking at most a few seconds.

### G.1 Failure Case Analysis

To better understand whether to failure cases in the perturbation experiments from Section 3.2 stemmed from decomposition or recovery issues, we report the per-example reconstruction loss in Table 10 and histograms of pseudo-Fisher vector absolute cosine-similarities in Table 11. Recall that the failure cases were gradient clustering on YAT and NPEFF on CLINC150 and TriviaQA. We do not see substantially higher reconstruction losses for these cases that would indicate a failed decomposition. The pseudo-Fisher vector cosine similarities had similar distributions for both the failed NPEFF cases and the corresponding successful G-NPEFF cases. Gradient clusters on YAT had a few more highly similar clusters than the corresponding NPEFF and G-NPEFF decompositions on YAT. However, gradient clusters tended to have more of these highly similar clusters than the corresponding NPEFF variants even when the perturbation experiments succeeded.

Table 12: Percentage of matching components from NPEFF decomposition 2 with the same tuning as their match from NPEFF decomposition 1 for SST2.

| NPEFF Comps 1 | NPEFF Comps 2 | Percent of Tuned Matches |
| --- | --- | --- |
| 32 | 64 | 81.3 |
| 32 | 128 | 89.0 |
| 32 | 256 | 85.6 |
| 32 | 512 | 83.1 |
| 64 | 128 | 91.2 |
| 64 | 256 | 88.0 |
| 64 | 512 | 84.0 |
| 128 | 256 | 86.9 |
| 128 | 512 | 83.1 |
| 256 | 512 | 87.1 |

Table 13: Percentage of matching components from NPEFF decomposition 2 with the same tuning as their match from NPEFF decomposition 1 for CLINC150.

| NPEFF Comps 1 | NPEFF Comps 2 | Percent of Tuned Matches |
| --- | --- | --- |
| 32 | 64 | 44.4 |
| 32 | 128 | 17.6 |
| 32 | 256 | 31.4 |
| 32 | 512 | 26.9 |
| 64 | 128 | 60.6 |
| 64 | 256 | 46.6 |
| 64 | 512 | 43.2 |
| 128 | 256 | 64.4 |
| 128 | 512 | 56.9 |
| 256 | 512 | 67.4 |

## H  Number of Components Ablation Full Breakdown

A breakdown of fraction of matching components for NPEFF decompositions with preserved tunings with varying number of components is provided in Table 12 for SST2 and Table 13 for CLINC150.

## I  SAE Training Details

As is common practice, we constrain rows of the encoder and columns of the decoder to have unit L2 norm, which causes these parameters to lie on a manifold. Like Bricken et al. (2023b), we project gradients onto the tangent space of this manifold before passing them to Adam. Following Gao et al. (2024), we initialize the decoder to the transpose of the encoder. We did not do anything else to mitigate the issue of "dead" latents during training since this was not a major issue for us. We normalized per-token activations to unit L2 norm before passing them to the SAE. We also did this when computing top examples for each SAE component. In all of our experiments, we used a learning rate of 1e-3. We used a batch size of 4096 activations and trained for 51,200 batches.

## J  Component Top Examples

Top examples of random components are presented in Figure 4 for SST2, in Figure 5 for YAT, in Figure 6 for CLINC150, and Figure 7 for TriviaQA.

**Component 1**

devastation

death

suffered

be killed

fatal ailments

**Component 2**

very funny romantic comedy

delightful romantic comedy

enjoyable comedy

heartfelt comedy

delightful comedy

**Component 3**

a fascinating glimpse of urban life and the class warfare that embroils two young men

that presents a fascinating glimpse of urban life and the class warfare that embroils two young men

presents a fascinating glimpse of urban life and the class warfare that embroils two young men

a lively and engaging examination of how similar obsessions can dominate a family .

a lively and engaging examination of how similar obsessions can dominate a family

**Component 4**

its 112-minute length

notice the 129-minute running time

seems twice as long as its 83 minutes

its three-hour running time plays closer to two .

runs 163 minutes

**Component 5**

quite possibly the sturdiest example yet

smarter and more diabolical

far more entertaining than i had expected

makes oliver far more interesting

would seem to be surefire casting

**Component 6**

is the case of a pregnant premise being wasted by a script that takes few chances and manages to insult the intelligence of everyone in the audience

to it – as if the director is trying to dupe the viewer into taking it all as very important simply because the movie is ugly to look at and not a hollywood product

be a movie that ends up slapping its target audience in the face by shooting itself in the foot

to dupe the viewer into taking it all as very important simply because the movie is ugly to look at and not a hollywood product

to make you feel guilty about ignoring what the filmmakers clearly believe

**Component 7**

leaves you wanting more

appetizer that leaves you wanting more

filled with raw emotions

really does feel like a short stretched out to feature length .

makes two hours feel like four .

**Component 8**

a tiresome cliché

to many clichés

the clumsy cliché

fails to keep it up and settles into clichés

a cliché

**Component 9**

which half of dragonfly is worse : the part where nothing 's happening , or the part where something 's happening

in a doctor 's office , emergency room , hospital bed or insurance company office

do n't know why steven seagal is considered a star , nor why he keeps being cast in action films when none of them are ever any good or make any money

does n't understand that the idea of exploiting molestation for laughs is funny , not actually exploiting it yourself

how inept is serving sara ?

**Component 10**

promising

inspires

exciting

exciting

powerful

Figure 4: Top examples of random components from NPEFF decompositions for SST2 in Section 3.1.

**Component 1**
Which sport is better baseball or basketball, and tell me which player is best at that sport.?

What is the most popular sport in the world as a whole: Soccer or American Football?

Which sport is better, baseball or basketball, and tell me which player is best at that sport.?

Which sport is better, baseball or basketball, and tell me which player is best at that sport.?

Basketball trivia. Which coach leads the NBA in assists in a game?

**Component 2**
What movies should I rent this weekend?

What is the single funniest scene in a movie you've ever seen?

What's the far best movie u seen of all time?

What is the funniest song you've ever heard?

What is the funniest movie dialogue you have ever liked?

**Component 3**
how to write a recommendation letter?

Where can I find good examples of cover letters (free)?

What are résumés supposed to look like?

How do you write a bibliography??

how do write a good resignation letter?

**Component 4**
how do u make a good paper airplane?

if someone killed your dog wat would u do!?

whats the trick to david blaines levitation?

if your **** doesnt grow how do you make it grow?

How do you stopyour mom from watching soap operas?

**Component 5**
What is better and can help to penis enlargement, brief or boxer shorts?

I'm looking for adult fleece loungewear.?

what is a good paintball gun for a begginner?

i am looking for a used 3 three wheel bicycle?

Where can I purchase liquid chalk? I live in Vista,CA.?

**Component 6**
I want to know who sing a christian song that has in its lyrics "when you've lost your faith Borrow mine.

DOES ANYONE KNOW WHERE I CAN FIND INFO ON PREP OR PRIVATE SCHOOLS IN THE BRONX THAT AREN'T CATHOLIC SCHOOLS?

Valentines poems for moms?

Looking for a good Ballet School for my toddler in San Diego, CA?

what does st. patrick's day mean?

**Component 7**
what's the best / worst thing that happened to you this year?

what is the worst part about America?

is it bad to care about one of my jobs and not the other one?

what drives you crazy?

what kind of disabilities are there?

**Component 8**
I have marked e-mail spam by mistake. How do I retrieve the addresses so they are not considered spam?

When I click on a mailto: link, Outlook opens Word as my e-mail editor. How do I change my default editor?

The mail I send to Yahoo from my office email gets treated as spam in Yahoo. How do I correct this?

My sister and I use to email. She has since passed away. Is there a way to retrieve deleted emails?

When using Yahoo Messenger 7.0, how do you delete custom Status Messages (aka Away Messages)?

**Component 9**
What is black hole?

What is black hole?

What exactly is a black hole?

what is black hole?

what is black hole?

**Component 10**
what's it ?

what do i do ?

whats on your mind?

what should I do?

what do you think about this?

Figure 5: Top examples of random components from NPEFF decompositions for YAT in Section 3.1.

**Component 1**

i need to know how many vacation days i have

i need to know the number of days off i have taken at this point

i need to know how many of my days off i have used at this point

i need to know how many days off i have used so far

i need to know how many days i toof off

**Component 2**

i can't locate my mastercard and i want to report it as lost, please

can i get some more checkbooks sent to me, please

where can i get the form i need to do my taxes, please

how can i apply for a mastercard, please

restore your original settings, please

**Component 3**

i'm thankful for your help

i thank you

thank you for your help

i would like to thank you ai

thank you very much for the assistance

**Component 4**

where should i look for my w-2

where can i find my w-2

where do i go to get my w-2

where can i get my w-2

can you locate my w-2

**Component 5**

can you not talk so fast

you talk too slow

can you please talk slower

can you please not talk so fast

can you please talk faster

**Component 6**

does longhorn steakhouse have good reviews

does olive garden have good customer reviews

does outback steakhouse have good reviews

does olive garden have good reviews

does pizza hut have good reviews

**Component 7**

how much money do i pay in taxes

how much do i pay in my taxes

how much will i pay in state taxes

how much do i pay in taxes every year

how much do i pay in taxes

**Component 8**

i really need to get a volkswagen car rental for march 5th to march 8th in phoenix

i want to rent a bmw suv for dallas from march 2 to 6th

i need to let my bank know i will be in america from april to may

i need to rent an suv in charlestown for the first week in june who do you suggest

i want a bmw suv for march 2 to 6th in dallas

**Component 9**

i need a recipe for chili

find a recipe for baked ziti

find me a recipe for chili

i need a recipe for chicken cordon bleu

find a recipe for hamburgers

**Component 10**

can you tell me if i will have any transactions fees for using my discover card in turkey

how much is 1 share of aapl

how much is the exchange between usd and euros

whats the current exchange rate between usd and eur

i suspect fraudulent transaction

Figure 6: Top examples of random components from NPEFF decompositions for CLINC150 in Section 3.1.

**Component 1**
In Judaism, what is a 'Kever'?
In heraldry, what is a wyvern?
What is a Stiffkey (or Stewkey) Blue?
In CB jargon what is a bone box?
What is a 'tercel' ('tiercel' in the USA)?

**Component 2**
For her performance in which film did Meryl Streep win the Best Actress 'Oscar' earlier this year?
For her performance in which film did Meryl Streep win the Best Actress 'Oscar' earlier this year?
For his role in which film did Al Pacino win the Oscar for best actor in 1992?
For her role in which film did Jane Darwell win the 'Oscar' for Best Supporting Actress in 1940 when aged 61?
For his role in which film did Al Pacino win the Oscar for best actor in 1992?

**Component 3**
"The names of how many US states begin with ""New""?"
"The names of how many US states begin with ""M""?"
"The names of how many US states begin with the letter ""I""?"
"The names of how many US states begin with ""New""?"
"The names of how many US states begin with ""M""?"

**Component 4**
Which famous actor was born in Beirut in 1964?
Which theatre critic devised and produced the erotic revue 'Oh! Calcutta!' in 1969?
Which poet died of septicaemia in the Aegean Sea in 1915?
Which theatre critic devised and produced the erotic revue 'Oh! Calcutta!' in 1969?
Which famous author spent 5 years in the 1920's as a police officer in Burma?

**Component 5**
A rat can survive longer without water than a camel?
A rat can survive longer without water than a camel?
A Wayzgoose is an annual outing or party in which industry?
A system known as SCOOT (Split Cycle Offset Optimisation Technique) is used to control our travel movements in what way?
A Sugar anniversary celebrates how many years of marriage?

**Component 6**
Which film tells of the exploits of singer Deco Duffe?
Which film tells of the exploits of singer Deco Duffe?
Which sitcom with Vickie Lawrence was a spin-off from the Carol Burnett Show?
Which sitcom with Vickie Lawrence was a spin-off from the Carol Burnett Show?
Which show was based on the autobiography of Gypsy Rose Lee?

**Component 6**
Who was Henry VIII's third wife?
Who was Henry VIII's third wife?
British monarch Henry VIII married which of his wives in 1540?
British monarch Henry VIII married which of his wives in 1540?
Which of Henry VIII's wives was the mother of Mary I?

**Component 7**
"Which product was advertised on TV with the slogan ""Good to the last drop""?"
"Which product was advertised with the slogan ""Good to the last drop""?"
"Which product was advertised with the slogan, ""Forces grey out, forces white in""?"
"Which famous product was advertised on TV with the words ""Cleans and polishes in one go""?"
"Which famous product was advertised on TV with the words ""it won't let you down""?"

**Component 8**
How many apprentice boys shut the gates of Derry in December 1688 leading to the siege of the city?
How many theses did Martin Luther post on the door of the Castle Church of Wittenberg in October 1517?
How many Victoria Crosses were won at Rorke's Drift in 1879?
How many theses were nailed to a church door by Martin Luther in 1517 (generally accepted to be the Castle Church in Wittenberg, Germany, on All Saints Eve, 31 October)?
How many prisoners were locked in the Bastille in Paris when it was stormed by the people in 1789?

**Component 9**
Which Shakespeare play is mainly set in a forest outside Athens?
Which Shakespeare play is set in Illyria?
Which Shakespeare play is mainly set in a forest outside Athens?
Which Shakespeare play has the siege of Troy as its setting?
Which Shakespeare play has the siege of Troy as its setting?

**Component 10**
Which gas forms 80% of Earth's atmosphere?
Which gas forms 80% of Earth's atmosphere?
Which gas forms approximately 1% of the atmosphere?
Which gas forms approximately 1% of the atmosphere?
Which gas forms about 78% of the Earth's atmosphere?

Figure 7: Top examples of random components from NPEFF decompositions for TriviaQA in Section 3.1.

