# OpenReview forum: "Uncovering Language Model Processing Strategies with Non-Negative Per-Example Fisher Factorization"
_TMLR — Rejected by TMLR_

### Review · Reviewer_EwL6 · 2025-08-31

**Summary Of Contributions:**

The paper introduces NPEFF, an interpretability method that decomposes per-example Fisher (PEF) matrices into a non-negative combination of rank-1 positive semi-definite components. Each component is intended to reflect a behavior factor used by a language model on a given input. The work also proposes G-NPEFF, a lower-cost variant that applies the same factorization to gradients of the predicted class.

**Audience:**

Yes

**Audience Explanation:**

Yes. The paper advances unsupervised behavior discovery for language models using a principled object (per-example Fisher) and connects it to both interpretability and controllability.

**Broader Impact Concerns:**

The work enables targeted manipulation of model behaviors by constructing parameter perturbations aligned with discovered factors. This can support safety testing, but it can also be used to degrade desirable behaviors or to induce undesired behaviors in deployed models. The paper should include a short discussion of dual-use risks, appropriate safeguards for releasing perturbation code, and the ethical implications of extracting and steering behavior factors in fine-tuned or proprietary models.

**Claims And Evidence:**

Yes

**Claims Explanation:**

Mostly yes. The qualitative top-example inspections and the human study on TriviaQA support interpretability claims. The label-not-prediction tuning metric captures expected properties of polygenic factors and consistently favors NPEFF over baselines as the class count grows.

**Requested Changes:**

First, please analyze the failure cases in the perturbation study in more depth. For each case, report the reconstruction quality of projected versus recovered parameter-space pseudo-Fisher vectors, the cosine similarity distribution among components, and the fraction of explained Frobenius norm per example. This will help separate decomposition issues from recovery issues.

Second, expand the sensitivity analysis. In addition to the ablations on projection size, SVD rank, and expectation projections, include variance across at least three random seeds for the decomposition, with confidence intervals for the key metrics (prediction-tuned, label-not-prediction tuned, and perturbation KL ratios). Provide a practical recipe for choosing the number of components, for example a plot of validation reconstruction error and the stability of component tunings as the component count increases.

---

> ### Author Response · Authors · 2025-09-21
> **Author Rebuttal**
>
> Thank you for your review.
>
> > please analyze the failure cases in the perturbation study in more depth.
>
> As requested, we have added a “Failure Case Analysis” section to the appendix that is referenced in the main perturbations section. We include a table of per-example reconstruction losses (i.e. per-example squared Frobenius/L2 distance) for all of the decompositions. We have also included a table of histograms of absolute cosine-similarities of pseudo-Fisher vectors for all of the decompositions. From these results, we observe that there is not anything particularly suspect about the cases that failed. This supports our assertion that the issue was likely a recovery issue rather than a decomposition issue.
>
> > reconstruction quality of projected versus recovered parameter-space pseudo-Fisher vectors
>
> Because we do not have access to the ground-truth parameter-space pseudo-Fisher vectors, we cannot report the reconstruction quality of the recovered pseudo-Fisher vectors. Was there a different notion of reconstruction quality that you want us to report?
>
> > include variance across at least three random seeds for the decomposition
>
> We have added a “Decomposition Random Seed” paragraph to the Ablations section where we report prediction-tuned percentage, LnP-tuned percentage, perturbation KL ratio, and perturbation norm ratio for 3 different random seeds. Across the seeds, we see only a minor variation for the fractions of tuned components and the selectivity of perturbations. This indicates that the properties of an NPEFF decomposition is robust to the random seed
> used to initialize it.
>
> > Provide a practical recipe for choosing the number of components, for example a plot of validation reconstruction error and the stability of component tunings as the component count increases.
>
> We have added to the “Number of Components” paragraph of the Ablations section with a table of per-example held-in reconstruction losses, per-example held-out reconstruction losses, and component tuning percentages on the held-out set for SST2 and CLINC150. Overall, the desired granularity of the uncovered factors plays the biggest role in the choice of the number of components in the NPEFF decomposition. Even for a relatively high number of components such as around 10% of the number of examples for CLINC150, we do not observe substantial overfitting.
>
> > Broader impact concerns
>
> We have added a broader impact statement. It reads as:
> “NPEFF provides a means to decompose the behavior of a language model into factors and a means to relate those factors to directions in parameter space. This could enable directed modification of model behaviors. While this can be beneficial if positive behaviors are amplified and negative behaviors are suppressed, this could be harmful if used in the opposite way. While we release our perturbation code, it only allows for disruption of behaviors and is insufficient for more sophisticated modification of model behavior.”

---

### Review · Reviewer_H6iD · 2025-09-02

**Summary Of Contributions:**

This paper introduces NPEFF (Non-Negative Per-Example Fisher Factorization), a new way to look inside large language models and understand how they  make predictions. Unlike older methods that assume each example is driven by just one factor, NPEFF can capture cases where multiple factors work together. It does this by breaking down special matrices called per-example Fishers into simpler pieces, using tricks like random projections to keep things efficient. The authors show that NPEFF finds more meaningful patterns than existing approaches, lets you “poke” the model by disrupting specific strategies, and even sheds light on how in-context learning mostly re-weights behaviors the model already has rather than inventing new ones.

**Audience:**

Yes

**Audience Explanation:**

1. This paper provides an interesting approach to language model interpretability. Given that these models are being increasingly deployed in the real-world, it is important to gain insight into their inner behaviors. I would suggest the authors provide more discussion on how would the interpretability results help make language models better.

2. The authors should also discuss the motivation for the proposed approach at the start of Section 2 and highlight the advantages and possible disadvantages as compared to other interpretability methods.

**Broader Impact Concerns:**

I do not have concerns regarding broader impacts of this paper.

**Claims And Evidence:**

Yes

**Claims Explanation:**

I think the authors did a good job of evaluating their proposed approach. The authors run experiments across multiple benchmarks (SST2, Yahoo Answers, CLINC150, and TriviaQA) and consistently compare NPEFF against strong baselines such as gradient clustering and sparse autoencoders, showing that NPEFF better captures polygenic behaviors.

However, I think the paper miss other important baselines for interpreting language model behaviors. For example, [1] used influence function approximations to study the effect of training sequences. Also it would be great to extend the evaluation to datasets beyond simple question answering datasets.


[1] Studying Large Language Model Generalization with Influence Functions. Grosse, 2023.

**Requested Changes:**

The authors should discuss more on the novelty contributions of the proposed approach as compared to baselines (gradient clustering and SAEs), and also other approaches, e.g., [1].



[1] Studying Large Language Model Generalization with Influence Functions. Grosse, 2023.

---

> ### Author Response · Authors · 2025-09-21
> **Author Rebuttal**
>
> Thank you for your review.
>
> > Influence functions
>
> The main reason we did not include influence functions as a baseline in the paper was that it does not explain model predictions in terms of broad interpretable factors. Instead, it explains the model predictions in terms of effects of specific training examples. Since our analysis focused on the properties of the extracted factors, they would not have been possible to perform on influence functions. We did, however, include influence functions in the related work. We have added a sentence explaining why we do not include influence functions as a baseline in the section where we introduce the baselines.
>
> > I would suggest the authors provide more discussion on how would the interpretability results help make language models better.
>
> We have added a paragraph in the Discussion section titled “Improving language models” discussing this. It reads as:
> “Though we have mostly focused in this work on introducing and evaluating NPEFF as an interpretability method, the potential exists to use it to improve language models. Examining component tunings could uncover beneficial or problematic behaviors. We also provide a means to produce the directions in parameter space corresponding to particular behaviors. These could be used as in our perturbation experiments to selectively disrupt particular behaviors. We leave development of more sophisticated methods to modify model behavior based on NPEFF to future work.”
>
> > The authors should also discuss the motivation for the proposed approach at the start of Section 2
>
> We have added a paragraph at the start of Section 2 summarizing and motivating our approach. It reads as:
> “NPEFF consists of two main stages: computation of PEFs and decomposition over a set of PEFs. We use PEFs to capture per-example processing since they relate perturbations in parameters to perturbations over the model's entire predictive distribution. We use a combination of low rank representations and random projections to make the storage of PEFs over many examples tractable. The decomposition stage aims to represent the PEFs as a non-negative combination of rank-1 positive semi-definite (PSD) matrices, which ensures that the reconstructions are PSD like the PEFs themselves. By allowing multiple factors to be assigned to each PEF, this decomposition respects the polygenicity of the underlying behavioral factors.”
>
> > highlight the advantages and possible disadvantages as compared to other interpretability methods.
>
> We have added a paragraph titled “Comparison to existing methods” to the Discussion section discussing this. We also compare NPEFF to influence functions there. It reads as:
> “The two main novelties introduced by NPEFF are using PEFs to characterize per-example processing and the decomposition via non-negative coefficients over rank-1 PSD matrices. Compared to gradient clustering (Michaud et al., 2023), NPEFF better captures the ground truth where multiple factors influence the model’s behavior on any particular example. Activation SAEs (Gao et al., 2024) provide an alternative view on model internals since activations capture what information is present at a particular location within the model but do a poor job at capturing the computation. Influence functions (Grosse et al., 2023) explain behavior in terms of influential training examples. Hence they cannot find behavioral factors like NPEFF. The main disadvantage of NPEFF the computational overhead associated with PEFs. We explored mitigating this by running NPEFF’s decomposition on gradients for G-NPEFF, but we found that G-NPEFF performed poorly at uncovering polygenic components as the number of classes increased.”
>
> > The authors should discuss more on the novelty contributions of the proposed approach as compared to baselines
>
> We do this in the paragraph titled “Comparison to existing methods” that we have added to the Discussion section.

---

### Review · Reviewer_JVD4 · 2025-09-08

**Summary Of Contributions:**

The paper addresses the issue of understanding how the decision is done by models (interpretability): instead of clustering gradients of the model authors propose to compute per example Fisher matrix and then find 1-rank matrices to have the best decomposition of these Fisher matrices for all examples, called . Authors show that these 1-rank matrices represent behaviour of the model including polygenic cases (when behavior is influenced by multiple factors from within the model). Authors also propose to use random projections to perform dimensionality reduction to speed up the method (and having CUDA kernels implementations for efficient computations) and show that this variant is performing similar if number of classes is small, but also this variant is working more robust for the perturbed model. Authors also investigate how ICL is working given new representation of the interpretable components: they conclude that much of the gains from ICL come from adapting to the specific presentation of the task rather than the model learning new behaviors from the context (for the case / data they considered). Empirical analysis is done on several text data with various numbers of classes for ~300M parameter LM.

**Strengths**
- well written paper
- interesting approach how to analyse model behaviour and get better decomposition on 1-rank representations
- extensive ablations and experimentation, design of analysis
- interesting results on in-context learning (ICL)
- efficient implementation with CUDA kernels and optimizations with random projections
- "Examining properties of NPEFF components, we saw that they corresponded to interpretable factors of behavior."

**Weaknesses**
- I would like to see runtime comparisons for different methods and code release for CUDA kernels which authors discuss
- I would like to see several LMs to be analysed - not only one small model, maybe look at least into 1/3B models if resources allow?
- Ablation on the number of components should be done (in my opinion) on the data with many classes (not 2 classes).
- There is no convergence guarantee, though authors discuss this issue in Appendix (This is minor, as I understand it may be infeasible to get any theoretical results. But it would be good to comment on the empirical results if there were any convergence issues).

**Audience:**

Yes

**Audience Explanation:**

The paper tries to tackle interpretability problems, also first interesting finding for ICL and what model is actually doing - this will be interesting to the broader community, and can be incorporated as a standard tool in model analysis.

**Broader Impact Concerns:**

The work is more theoretical method and empirical analysis and doesn't have any serious ethical implications right away.

**Claims And Evidence:**

Yes

**Claims Explanation:**

Overall the paper is clearly written, and the proposed method and idea are clear too. A bunch of pieces are reused, but all makes sense in the way the author put them together. Convergence is not proven however there is meaningful discussion in the Appendix. Authors provide fair and careful comparison with baseline (clustering of the grads), keeping the conditions consistent with the proposed method. Authors consider several datasets with various tasks and number of classes. Authors do different types of analysis including perturbation cases, ICL, and various ablations. Overall evaluation seems correct and solid, experimental design is meaningful and w/o any serious flaws, though for some places I have some questions and comments (see them below).

**Requested Changes:**

See weaknesses section above and please find below additional comments.

*Improving paper readability:*
- "why a language model generated" -> "why a language model generates"
- "we demonstrate that is possible" -> "we demonstrate that it is possible"
- "When construct the full PEF, when" -> "When construct the full PEF, we"
- "of each label labels" - I didn't get the correct text here

*Comments / questions:*
- In sec. 2.1.2 it is discussed rank reduction and it seems rank reduction is done along the classes axis, not the number of model parameters. This creates a huge confusion as later in the main experiments sections it is written that rank reduction is done per row, which corresponds to reduction along the model's parameters axis. This either leads to incorrect math or huge confusion during the rest of the paper reading (including Appendix derivations on how random projections change decomposition).
- "storage and handling of these gradients tractable" - I am not sure if this is really expensive as training models with such parameters is possible, thus it would be good to give some runtime estimation here. I am also not sure about the runtime for decomposition - good to know exact numbers too, as then the need for dim reduction is really solving the decomposition problem.
- Sec. 3.1 How do we use the number of examples of the same class? Do we depend on it and if yes, then how?
- Sec. 3.1 Why do you exclude layer norm and embedding parameters? Are they important / not important? maybe we should also exclude earlier layers of the network as it is less representative?
- "Gradient clustering (GC) performs k-means clustering on gradients of the log-probability of the predicted class/token for each example, which are the same gradients used by G-NPEFF" - not clear why this is optimal for baseline, maybe we need to consider raw grads?
- The whole subsection on "Verifying polygenicity" is not clear to me due to absence of definitions: what is polygenicity exactly, how do we measure it, what is prediction-tuned and label-tuned? While overall I can capture meaning, in general it is a very broad interpretation while you read the paragraph. Also how do you verify that component is prediction-tuned or label-tuned? Clear strong definitions with protocol on the verification will be helpful (even if there is no proof, or strong math derivations).
- "For most settings, the component top examples were significantly more affected by the perturbations than the random examples. This difference cannot be explained by component top examples simply being more sensitive to perturbations as indicated by the PEF norm ratios. These results indicate that the uncovered behavior factors play a genuine role in the model behavior." - can you explain this test? Why does what we observe lead to the conclusion exactly?
- Table 4: Can we say from this how much overparametrization is actually happening? I also wonder if we can skip earlier layers in the Fisher computation, do we get the same picture, meaning that earlier layers are not very important for interpretability?
- Ablation on number of components: did you try with other dataset as it seems weird to increase components for the data with 2 classes and it is obvious that it will be splitting the class into subclasses.
- Figure 2: can we try larger values to see saturation? Right now all border values used are the best, so it is not clear what the general pattern we have.
- Appendix A: for both paragraphs it will be great to have figures with visualizations, otherwise a bit hard to follow exact derivations in the text.
- The expression "rank-3 tensor" in the Appendix is confusing - it should be 3-dim tensor, not rank as rank is associated (also used in the whole paper before) as the matrix rank.

Please comment on the above questions and comments, and please include in the revision either fixes or clarifications for the raised points.

---

> ### Author Response · Authors · 2025-09-21
> **Author Rebuttal**
>
> > I would like to see runtime comparisons for different methods and code release for CUDA kernels which authors discuss
>
> We have included a section title “Runtime Information” in the appendix containing runtime comparisons between the methods. The CUDA kernels are contained in the supplemental material. We will release the code on github once the paper is deanonymized.
>
> > I would like to see several LMs to be analysed - not only one small model, maybe look at least into 1/3B models if resources allow?
>
> We have added a section title Experiments on SmolLM2-1.7B to the appendix containing component tunings for SmolLM2-1.7B, a 1.7B parameter model for SST2 and YAT. The results there mirror those in the main text: NPEFF component tunings are most consistent with recovery of polygenic factors with a high fraction of both prediction-tuned and non-prediction-tuned components and with the highest fraction of LnP-tuned components.
>
>
> > it would be good to comment on the empirical results if there were any convergence issues
>
> We have added “In practice, we did not encounter any issues with convergence given a long enough $H$-only update stage and a sufficiently low learning rate.” to the Convergence section in the appendix.
>
> > In sec. 2.1.2 it is discussed rank reduction…
>
> The rank reduction discussed in Section 2.1.2 and the random projections discussed in 2.2.1 are separate dimension reductions applied to the columns and rows of $G(x)$, respectively. We have added “...by decreasing its number of columns” to Section 2.1.2 and “Note that this dimensionality reduction is separate to the SVD-based approach in \cref{sec:rank_reduction} that reduces the number of \emph{columns} of $G(\x)$.” to Section 2.2.1 to help clarify that these are separate dimensionality reductions.
>
> > “storage and handling of these gradients tractable”
>
> Storing the gradients for a single example is indeed tractable. However, we need to store and process gradients across many examples, which would become intractable for the number of examples used in this paper. For example, storing gradients for 100K examples for the 360M parameter model would require storage of around 36T floating point values. We have changed this to “storage and handling of these gradients across many examples tractable” to help clarify this.
>
> > “Sec. 3.1 How do we use the number of examples of the same class? Do we depend on it and if yes, then how?”
>
> The computation of PEFs and decompositions are completely independent of the class labels, so the number of examples of the same class has absolutely no direct bearing on the decompositions. For tuning, we call a component label-tuned if all its 16 top examples (i.e. those with the largest coefficients) have the same ground truth label. Determining component tunings matters for analysis of a decomposition and not the decomposition itself.
>
> > Sec. 3.1 Why do you exclude layer norm and embedding parameters? Are they important / not important? maybe we should also exclude earlier layers of the network as it is less representative?
>
> We did this to focus on model internals as was done in Marks et al. (2024). We have not experimented with different forms of parameter selection, but we suspect that this choice does not significantly influence the components returned by NPEFF. It also has the advantage of slightly speeding up the random projections. We have added “...to focus on the processing done by the model's internals. We leave further exploration of parameter selection to future work.” to help clarify this.
>
> > "Gradient clustering (GC) performs k-means clustering on gradients of the log-probability of the predicted class/token for each example, which are the same gradients used by G-NPEFF" - not clear why this is optimal for baseline, maybe we need to consider raw grads?
>
> Previous works used the gradient of the loss for clustering, but they were restricted to examples where the model made the correct prediction. The gradient of the log-probability of the predicted class/token coincides with this gradient on those examples. The other methods do not require ground truth labels, so we used this to make GC compatible with these requirements. Furthermore, it is not clear if using the loss gradient would capture much model processing when the model puts low weight on the correction prediction/token. We have added the sentence “Like the other methods, this enables GC to not require ground truth labels and coincides with the gradient of the loss when restricted to examples where the model makes the correct prediction as was done in Michaud et al. (2023).” to help clarify this.

---

> > ### Author Response · Authors · 2025-09-21
> > **Author Rebuttal (part 2)**
> >
> > > The whole subsection on "Verifying polygenicity" is not clear to me due to absence of definitions: what is polygenicity exactly, how do we measure it, what is prediction-tuned and label-tuned? While overall I can capture meaning, in general it is a very broad interpretation while you read the paragraph. Also how do you verify that component is prediction-tuned or label-tuned?
> >
> > We have “We can perform an automated analysis by considering a component as tuned if all of its top 16 examples had the same prediction or ground truth label, if present.” in the paper defining prediction/label-tuned.
> >
> > Polygenic behaviors are those influenced by multiple factors. We have added “Recall that polygenic behaviors are influenced by multiple factors.” to this section to be more explicit about this.
> >
> > > "For most settings, the component top examples were significantly more affected by the perturbations than the random examples. This difference cannot be explained by component top examples simply being more sensitive to perturbations as indicated by the PEF norm ratios. These results indicate that the uncovered behavior factors play a genuine role in the model behavior." - can you explain this test? Why does what we observe lead to the conclusion exactly?
> >
> > Under the hypothesis that the factors uncovered by NPEFF are genuine (i.e. correspond to factors used by the model that affect its behavior), then perturbing the parameters responsible for that factor should have a large effect on the model’s predictions on examples where it uses the factor and small effect otherwise. We use KL-divergence of the perturbed model’s predictions from the original model’s predictions to measure how much the perturbation affected the model on a particular example. We use the ratio of the average KL-divergence for the model’s top examples (i.e. where the factor is used) to average KL-divergence for random examples (i.e. where the factor is not likely to be used) to measure the selectivity of the perturbation.
> >
> > One caveat is that the model’s behavior on certain examples can be more sensitive to perturbations in general. Hence we want to make sure that a high KL ratio for a component isn’t just a result of its top examples being more sensitive to perturbations in general. We use the PEF Frobenius norm as a proxy for this (See equation 2 for why KL-divergence scales with the PEF Frobenius norm), so we report the ratio of average PEF norms for the component top examples to the average for random examples. Since this ratio was always less than 1, component top examples were not more generally sensitive to perturbations, so this does not explain why the KL-ratios were significantly larger than 1. Hence, the reason for the large KL-ratio was that the perturbations selectively disrupted a factor important to the component top examples and not random examples.
> >
> > We have added “... since we selectively disrupted the model's behavior on examples where a particular factor was deemed important.” to help clarify this.
> >
> > > Table 4: Can we say from this how much overparametrization is actually happening?
> >
> > We don’t think it is possible to derive many conclusions about the overparameterization of the original model from this table. It is likely that far less information is needed for a “good enough” snapshot of model processing for the purposes of decomposition than for the purposes of actually recreating the processing.
> >
> > > Ablation on number of components: did you try with other dataset as it seems weird to increase components for the data with 2 classes and it is obvious that it will be splitting the class into subclasses.
> >
> > We have added results for CLINC150 where we change the number of components to that section. We get a value of 46.0% of fine-grained splits having the same prediction-tuning as their coarse-grained “parent”, which is significant but lower than the 85.9% we have for SST2. We added “The lower value for CLINC150 might come from coarse-grained components actually being tuned to multiple predictions but biased towards a specific prediction.
> > These would register as tuned to a single prediction but would have some fine-grained splits tuned to a different predictions.” to that section to explain why it would be lower for a task with more classes.
> >
> > > Figure 2: can we try larger values to see saturation? Right now all border values used are the best, so it is not clear what the general pattern we have.
> >
> > The SVD-reduced rank is bounded from above by the number of projections used when approximating the expectation. Thus we cannot use larger values without exceeding the number of projections. The general pattern is that increasing the rank leads to a decomposition more similar to the non-rank reduced case. We have added the sentence “We observe the general pattern of higher SVD-reduced ranks leading to decompositions more similar to the non-reduced rank case.” to help clarify this.
> >
> > > "rank-3 tensor"
> > We have changed “rank-3” to “3-dim” in this case.

---

> > > ### Comment · Reviewer_JVD4 · 2025-09-23
> > > **Reply**
> > >
> > > Dear Authors,
> > >
> > > Thanks for all clarifications and additional experiments (another larger LM model and another dataset). I am happy with your responses. Some specific comments:
> > >
> > > > Storing the gradients for a single example is indeed tractable. However, we need to store and process gradients across many examples, which would become intractable for the number of examples used in this paper. For example, storing gradients for 100K examples for the 360M parameter model would require storage of around 36T floating point values. We have changed this to “storage and handling of these gradients across many examples tractable” to help clarify this.
> > >
> > > Ahh, got it. So the problem is not batched regime as in training and compute grads for many-many samples. Thanks for clarification!
> > >
> > > > We have “We can perform an automated analysis by considering a component as tuned if all of its top 16 examples had the same prediction or ground truth label, if present.” in the paper defining prediction/label-tuned.
> > >
> > > I think I didn’t get exactly what it means for component to be tuned. I think now I got what you mean.
> > >
> > > > The SVD-reduced rank is bounded from above by the number of projections used when approximating the expectation. Thus we cannot use larger values without exceeding the number of projections. The general pattern is that increasing the rank leads to a decomposition more similar to the non-rank reduced case. We have added the sentence “We observe the general pattern of higher SVD-reduced ranks leading to decompositions more similar to the non-reduced rank case.” to help clarify this.
> > >
> > > Got it! Often I add the topline in this case pointing the limit and what happens when we do approximation. But I am good with the way you added clarification!
> > >
> > > I am checking the final revision, will update asap. By any chance, could you make red-colored pieces you changed? If not it is ok too, just need a bit more time to do the final proof read.
> > >
> > > Thanks,
> > >
> > > Reviewer.

---

> > > > ### Author Response · Authors · 2025-09-23
> > > > **Reply**
> > > >
> > > > Thank you for your reply.
> > > >
> > > > We have updated the revision so that the changed pieces are red.

---

> > > > > ### Comment · Reviewer_JVD4 · 2025-09-24
> > > > > **Reply**
> > > > >
> > > > > Dear Authors,
> > > > >
> > > > > Thank you for the revision with changes marked as red. I did a pass, and happy with the version.
> > > > >
> > > > > Maybe last question I have is when you do SVD for reduction across column (so thus across classes) - how would later interpret results given that now you don't have per class information - but rather it will have some other general classes information?
> > > > >
> > > > > Thanks,
> > > > >
> > > > > Reviewer.

---

> > > > > > ### Author Response · Authors · 2025-09-25
> > > > > > **Reply**
> > > > > >
> > > > > > This does not change the substance of your question, but we accidently swapped "columns" for "rows" and vice-versa when discussing the LRM-PEF in our initial changes and rebuttal. This was just a typo (i.e. everything still stands if you swap "column" for "row" and vice-versa), so we have corrected this in our latest revision. Columns and rows are used correctly in the rest of the paper.
> > > > > >
> > > > > > More precisely, we changed the "...by decreasing its number of columns." to "...by decreasing its number of rows." in Section 2.1.2, and we changed "...to the rows of $G(x)$. Note that this dimensionality reduction is separate to the SVD-based approach in Section 2.1.2 that reduces the number of columns of $G(x)$." to "...to each row of $G(x)$, thus reducing its number of columns. Note that this dimensionality reduction is separate to the SVD-based approach in Section 2.1.2 that reduces the number of rows of $G(x)$." in Section 2.2.1.
> > > > > >
> > > > > > Apologies for the confusion.
> > > > > >
> > > > > >
> > > > > > To answer your question, we do not make use of rows of the LRM-PEF $G$ corresponding to classes since we only interact with it via the PEF $F=G^TG$, which sums over that dimension. When we use class information for determining prediction-tuning and label-tuning, this information is provided externally to what is present in the PEFs. More precisely when determining tuning for a component, we take as input its coefficient vector over examples and a vector of predictions or labels over examples. The predictions come from taking the argmax of $p_\theta(y|x)$ for each example, which is stored separately from the PEFs. The labels come from taking and storing the ground-truth labels for the examples. We further note that predictions and labels are used only for analysis given a decomposition and do not directly influence the decomposition itself.
> > > > > >
> > > > > > We have added the sentence "Note that we do not make use of the fact that rows of the LRM-PEF $G(x)$ correspond to categories since we only interact with it via the PEF $F(x) = G(x)^TG(x)$." to the end of Section 2.1 to help clarify this.

---

### Decision · Action_Editor_Kqik · 2026-01-04

**Recommendation:** Reject

**Additional Comments:**

Revisions checklist:
- Stabilize the perturbation/recovery pipeline or substantially strengthen the evidence around it,
 either fix the CLINC150/TriviaQA NPEFF failures or provide a principled diagnosis + conditions under which recovery is reliable.
- Broaden evaluation of interpretability beyond a single human study setting (even small-scale replication on another task would help).

- Clarify practicality and reproducibility:
  - include stronger guidance on compute/memory scaling and what is required without custom kernels, and/or provide clean reference implementations; runtime comparisons already highlight a big gap vs GC.
   - tighten baselines / positioning, the paper contrasts mostly with gradient clustering and activation SAEs; add or more clearly justify omissions of other sensitivity/attribution families.

**Audience:**

Yes

**Audience Explanation:**

Yes, all three reviewers found the paper interesting.

**Claims And Evidence:**

No

**Claims Explanation:**

In its current form, I would suggest the paper is not yet suitable for TMLR. Although it looks promising,  it would require more than minor revision to meet typical TMLR expectations on robustness, evidence, and practicality.

The perturbation section contains notable failure cases (NPEFF on CLINC150 and TriviaQA), and the paper cannot clearly isolate whether the issue is (i) learned projected representations or (ii) compressed-sensing recovery. This makes the strongest claim on getting parameter directions that selectively affect a factor feel not yet reliably established.

Human evaluation is restricted to TriviaQA due to cost, which limits the generality of the “themes are detectable” claim.

While interesting and potentially publishable in TMLR,  I would expect major revision rather than minor, primarily because the paper’s strongest “mechanistic/causal” line (parameter-space recovery + selective perturbations) has unresolved failure modes, and the evaluation breadth is still limited.

**Resubmission Of Major Revision:**

The authors may consider submitting a major revision at a later time.